# FUNDAMENTAL LIMITS IN FORMAL VERIFICATION OF MESSAGE-PASSING NEURAL NETWORKS

**Marco Sälzer**
School of Electr. Eng. and Computer Science
University of Kassel, Germany
marco.saelzer@uni-kassel.de

**Martin Lange**
School of Electr. Eng. and Computer Science
University of Kassel, Germany
martin.lange@uni-kassel.de

## ABSTRACT

Output reachability and adversarial robustness are among the most relevant safety properties of neural networks. We show that in the context of Message Passing Neural Networks (MPNN), a common Graph Neural Network (GNN) model, formal verification is impossible. In particular, we show that output reachability of graph-classifier MPNN, working over graphs of unbounded, but finite size, non-trivial degree and sufficiently expressive node labels, cannot be verified formally: there is no algorithm that answers correctly (with yes or no), given an graph-classifier MPNN, whether there exists some valid input to the MPNN such that the corresponding output satisfies a given specification. However, we also show that output reachability and adversarial robustness of node-classifier MPNN can be verified formally when a limit on the degree of input graphs is given a priori. We discuss the implications of these results, for the purpose of obtaining a complete picture of the principle possibility to formally verify GNN, depending on the expressiveness of the involved GNN models and input-output specifications.

## 1 INTRODUCTION

The Graph Neural Network (GNN) framework, i.e. models that compute functions over graphs, has become a goto technique for learning tasks over structured data. This is not surprising since GNN application possibilities are enormous, ranging from natural sciences (Kipf et al. (2018); Fout et al. (2017)) over recommender systems (Fan et al. (2019)) to general knowledge graph applications which itself includes a broad range of applications (Zhou et al. (2020)). Naturally, the high interest in GNN and their broad range of applications including safety-critical ones, for instance in traffic situations, impose two necessities: first, a solid foundational theory of GNN is needed that describes possibilities and limits of GNN models. Second, methods for assessing the safety of GNN are needed, in the best case giving guarantees for certain safety properties.

Compared to the amount of work on performance improvement for GNN or the development of new model variants, the amount of work studying basic theoretical results about GNN is rather limited. Some general results have been obtained as follows: independently, Xu et al. (2019) and Morris et al. (2019) showed that GNN belonging to the model of *Message Passing Neural Networks* (MPNN) (Gilmer et al. (2017)) are non-universal in the sense that they cannot be trained to distinguish specific graph structures. Furthermore, both relate the expressiveness of MPNN to the Weisfeiler-Leman graph isomorphism test. This characterisation is thoroughly described and extended by Grohe (2021). Loukas (2020) showed that MPNN can be Turing universal under certain conditions and gave impossibility results of MPNN with restricted depth and width for solving certain graph problems.

Similarly, there is a lack of work regarding safety guarantees for GNN, or in other words work on *formal verification* of GNN. Research in this direction is almost exclusively concerned with certifying *adversarial robustness properties* (ARP) of node-classifying GNN (see Sect. 1.1 for details). There, usually considered ARP specify a set of valid inputs by giving a center graph and a bounded budget of allowed modifications and are satisfied by some GNN if all valid inputs are classified to the same, correct class. However, due to the nature of allowed modifications, these properties cover only local parts of the input space, namely neighbourhoods around a center graph.

This local notion of adversarial robustness is also common in formal verification of classical neural networks (NN). However, in NN verification, the absence of misbehaviour of a more global kind is adressed using so called *output reachability properties* (ORP) (Huang et al. (2020)). A common choice of ORP specifies a convex set of valid input vectors and a convex set of valid output vectors and is satisfied by some NN if there is a valid input that leads to a valid output. Thus, falsifying ORP, specifiying unwanted behaviour as valid outputs, guarantees the absence of respective misbehaviour regarding the set of valid inputs. To the best of our knowledge there currently is no research directly concerned with ORP of GNN.

This work adresses both of the above mentioned gaps: we present fundamental results regarding the (im-)possibility of formal verification of GNN. We prove that – in direct contrast to formal verification of NN – there are non-trivial classes of ORP and ARP used for MPNN graph classification, that cannot be verified formally. Namely, as soon as the chosen kind of input specifications allows for graphs of unbounded, but finite size, non-trivial degree and sufficiently expressive labels, formal verification is no longer automatically possible in the following sense: there is no algorithm that, given an MPNN and specifications of valid inputs and outputs, answers correctly (yes/no) whether some valid input is mapped to some (in-)valid output. Additionally, we show that ORP and ARP of MPNN used for node classification are formally verifiable as soon as the degree of valid input graphs is bounded. In the ARP case, this extends the previously known bounds.

The remaining part of this work is structured as follows: we give necessary definitions in Sect. 2 and a comprehensive overview of our results in Sect.3. In Sect. 4 and Sect. 5, we cover formal arguments, with purely technical parts outsourced to App. A and B. Finally, we discuss and evaluate our possibility and impossibility results in Sect.6.

## 1.1 RELATED WORK

This paper adresses fundamental questions regarding formal verification of adversarial robustness and output reachability of MPNN and GNN in general.

Günnemann (2022) presents a survey on recent developments in research on adversarial attack, defense and robustness of GNN. We recapitulate some categorizations made in the survey and rank the corresponding works in our results. First, according to Günnemann (2022) most work considers GNN used for node-classification (for example, Zügner et al. (2018); Dai et al. (2018); Wang et al. (2020); Wu et al. (2019)) and among such most common adversarial modifications are edge modifications of a fixed input graph (Zügner et al. (2018); Zügner & Günnemann (2019); Ma et al. (2020)), but also node injections or deletions are considered (Sun et al. (2020); Geisler et al. (2021)). In all cases, the amount of such discrete modifications is bounded, which means that the set of input graphs under consideration is finite and, thus, the maximal degree is bounded. Any argument for the possibility of formal verification derivable from these works is subsumed by Theorem 2 here.

Additionally, there is work considering label modifications (Zügner et al. (2018); Wu et al. (2019); Takahashi (2019)), but only in discrete settings or where allowed modifications are bounded by box constraints. Again, this is covered by Theorem 2. There is also work on adversarial robustness of graph-classifier GNN (Jin et al. (2020); Chen et al. (2020); Bojchevski et al. (2020)). In all cases, the considered set of input graphs is given by a bounded amount of structural pertubations to some center graph. Therefore, this is no contradiction to the result of Corollary 1 as the size of considered graphs is always bounded.

As stated above, to the best of our knowledge, there currently is no work directly concerned with output reachability of MPNN or GNN in general.

## 2 PRELIMINARIES

**Undirected, labeled graphs and trees.** A *graph* $\mathcal{G}$ is a triple $(\mathbb{V}, \mathbb{D}, L)$ where $\mathbb{V}$ is a finite set of nodes, $\mathbb{D} \subseteq V^2$ a symmetric set of edges and $L : V \to \mathbb{R}^n$ is a labeling function, assigning a vector to each node. We define the *neighbourhood* $\mathrm{Neigh}(v)$ of a node $v$ as the set $\{v' \mid (v, v') \in \mathbb{D}\}$. The *degree* of $\mathcal{G}$ is the minimal $d \in \mathbb{N}$ s.t. for all $v \in \mathbb{V}$ we have $|\mathrm{Neigh}(v)| \leq d$. If the degree of $\mathcal{G}$ is $d$ then $\mathcal{G}$ is also called a *d-graph*. A *tree* $\mathcal{B}$ is a graph with specified node $v_0$, called the *root*, denoted by $(\mathbb{V}, \mathbb{D}, L, v_0)$ and the following properties: $\mathbb{V} = \mathbb{V}_0 \cup \mathbb{V}_1 \cup \cdots \cup \mathbb{V}_k$ where $\mathbb{V}_0 = \{v_0\}$, all $\mathbb{V}_i$

are pairwise disjoint, and whenever $(v, v') \in \mathbb{D}$ and $v \in \mathbb{V}_i$ then $v' \in \mathbb{V}_{i+1}$ or vice-versa, and for each node $v \in \mathbb{V}_i, i \geq 1$ there is exactly one $v' \in \mathbb{V}_{i-1}$ such that $(v', v) \in \mathbb{D}$. We call $k$ the *depth* of graph $\mathcal{B}$. A *d-tree* is a $d$-graph that is a tree.

**Neural networks.**  We only consider classical feed-forward neural networks using ReLU activations given by $re(\mathbf{x}) = \max(0, \mathbf{x})$ across all layers and simply refer to these as *neural networks* (NN). We use relatively small NN as building blocks to describe the structure of more complex ones. We call these small NN *gadgets* and typically define a gadget by specifying its computed function. This way of defining a gadget is ambiguous as there could be several, even infinitely many NN computing the same function. An obvious candidate will usually be clear from context. Let $N$ be a NN. We call $N$ *positive* if for all inputs $\boldsymbol{x}$ we have $N(\boldsymbol{x}) \geq 0$. We call $N$ *upwards bounded* if there is $\hat{n}$ with $\hat{n} \in \mathbb{R}$ such that $N(\boldsymbol{x}) \leq \hat{n}$ for all inputs $\boldsymbol{x}$.

**Message passing neural networks.**  A *Message Passing Neural Network (MPNN)* Gilmer et al. (2017) consists of layers $l_1, \ldots, l_k$ followed by a readout layer $l_{\mathsf{read}}$, which gives the overall output of the MPNN. Each regular layer $l_i$ computes $l_i(\mathbf{x}, \mathbb{M}) = \mathrm{comb}_i(\mathbf{x}, \mathrm{agg}_i(\mathbb{M}))$ where $\mathbb{M}$ is a multiset, a usual set but with duplicates, $\mathrm{agg}_i$ an aggregation function, mapping a multiset of vectors onto a single vector, $\mathrm{comb}_i$ a combination function, mapping two vectors of same dimension to a single one. In combination, layers $l_1, \ldots, l_k$ map each node $v \in \mathbb{V}$ of a graph $\mathcal{G} = (\mathbb{V}, \mathbb{D}, L)$ to a vector $\boldsymbol{x}_v^k$ in the following, recursive way: $\boldsymbol{x}_v^0 = L(v)$ and $\boldsymbol{x}_v^i = l_i(\boldsymbol{x}_v^{i-1}, \mathbb{M}_v^{i-1})$ where $\mathbb{M}_v^{i-1}$ is the multiset of vectors $\boldsymbol{x}_{v'}^{i-1}$ of all neighbours $v' \in \mathrm{Neigh}(v)$. We distinguish two kinds of MPNN, based on the form of $l_{\mathsf{read}}$: the readout layer $l_{\mathsf{read}}$ of a *node-classifier MPNN* computes $l_{\mathsf{read}}(v, \mathbb{M}^k) = read(\boldsymbol{x}_v^k)$ where $v$ is some designated node, $\mathbb{M}^k$ is the multiset of all vectors $\boldsymbol{x}_v^k$ and $read$ maps a single vector onto a single vector. The readout layer of a *graph-classifier MPNN* computes $l_{\mathsf{read}}(\mathbb{M}^k) = read(\sum_{v \in \mathbb{V}} \boldsymbol{x}_v^k)$. We denote the application of a node-classifier MPNN $N$ to $\mathcal{G}$ and $v$ by $N(\mathcal{G}, v)$ and the application of a graph-classifier $N$ to $\mathcal{G}$ by $N(\mathcal{G})$. In this paper, we make common choices (cf. Gilmer et al. (2017); Barceló et al. (2020); Wu et al. (2021)) for the form of the aggregation, combination and readout parts: $\mathrm{agg}_i(\mathbb{M}) = \sum_{\boldsymbol{x} \in \mathbb{M}} \boldsymbol{x}$, $\mathrm{comb}_i(\mathbf{x}, \mathbb{M}) = N_i(\mathbf{x}, \mathrm{agg}_i(\mathbb{M}))$ where $N_i$ is a NN and $read(\mathbb{M}) = N_r(\sum_{\boldsymbol{x} \in \mathbb{M}} \boldsymbol{x})$ respectively $read(\mathbf{x}) = N_r(\mathbf{x})$ where, again, $N_r$ is a NN.

**Input and output specifications.**  An *input specification* over graphs (resp. pairs of graphs and nodes) $\varphi$ is some formula, set of constraints, listing etc. that defines a set of graphs (resp. pairs of graphs and nodes) $\mathbb{S}_\varphi$. If a graph $\mathcal{G}$ (resp. pair $(\mathcal{G}, v)$) is included in $\mathbb{S}_\varphi$ we say that it is *valid* regarding $\varphi$ or that it *satisfies* $\varphi$, written $\mathcal{G} \models \varphi$, resp. $(\mathcal{G}, v) \models \varphi$. Analogously, an *output specification* over vectors $\psi$ defines a set of valid or satisfying vectors of equal dimensions. Typically, we denote a set of input specifications by $\Phi$ and a set of output specifications by $\Psi$.

**Formal verification of adversarial robustness and output reachability properties.**  An *adversarial robustness property (ARP)* $P$ is a triple $P = (N, \varphi, \psi)$ where $N$ is a GNN, $\varphi$ some input specification and $\psi$ some output specification. We say that $P$ holds iff for all inputs $I \models \varphi$ we have $N(I) \models \psi$. We denote the set of all ARP with $\varphi \in \Phi$, $\psi \in \Psi$ and graph-classifier or node-classifier by $\mathrm{ARP}_{\mathsf{graph}}(\Phi, \Psi)$ respectively $\mathrm{ARP}_{\mathsf{node}}(\Phi, \Psi)$. We simply write $\mathrm{ARP}(\Phi, \Psi)$ when we make no distinction between graph- or node-classifiers. Analogously, an *output reachability property (ORP)* $Q$ is a triple $Q = (N, \varphi, \psi)$, which holds iff there is input $I \models \varphi$ such that $N(I) \models \psi$, and we define $\mathrm{ORP}_{\mathsf{graph}}(\Phi, \Psi)$, $\mathrm{ORP}_{\mathsf{node}}(\Phi, \Psi)$ and $\mathrm{ORP}(\Phi, \Psi)$ accordingly. Let $\mathcal{P}$ be a set of safety properties like $\mathrm{ARP}_{\mathsf{graph}}(\Phi, \Psi)$ or $\mathrm{ORP}_{\mathsf{node}}(\Phi, \Psi)$. We say that $\mathcal{P}$ is *formally verifiable*[1] if there is an algorithm $A$ satisfying two properties for all $P \in \mathcal{P}$: first, if $P$ holds then $A(P) = \top$ (completeness) and, second, if $A(P) = \top$ then $P$ holds (soundness).

## 3   OVERVIEW OF RESULTS

This work presents fundamental (im-)possibility results about formal verification of ARP and ORP of MPNN. Obviously, such results depend on the considered sets of specifications. All specification sets used in this work are described in detail in Appendix A.

---

[1]In other words, the problem of determining, given an MPNN $N$ and descriptions of valid inputs and outputs, whether the corresponding property holds, is *decidable*.

First, we establish a connection between ORP and ARP. For a set of output specifications $\Psi$ we define $\overline{\Psi} = \{\overline{\psi} \mid \psi \in \Psi\}$ where $\overline{\psi}$ defines exactly the set of vectors which do not satisfy $\psi$. We have that $\overline{\overline{\Psi}} = \Psi$.

**Lemma 1.** $\mathrm{ORP}(\Phi, \Psi)$ is formally verifiable if and only if $\mathrm{ARP}(\Phi, \overline{\Psi})$ is formally verifiable.

*Proof.* Note that the ARP $(N, \varphi, \psi)$ holds iff the ORP $(N, \varphi, \overline{\psi})$ does not hold. Hence, any algorithm for either of these can be transformed into an algorithm for the other problem by first complementing the output specification and flipping the yes/no answer in the end. □

This connection between ARP and ORP, while usually not given fomally, is folklore. For example, see the survey by Huang et al. (2020), describing the left-to-right direction of Lemma 1.

Our first core contribution is that, in contrast to verification of ORP of classical NN, there are natural sets of graph-classifier ORP, which cannot be verified formally. Let $\Phi_{\mathsf{unb}}$ be a set of graph specifications, allowing for unbounded, but finite size, non-trivial degree and sufficiently expressive labels, and let $\Psi_{\mathsf{eq}}$ be a set of vector specifications able to check if a certain dimension of a vector is equal to some fixed integer.

**Theorem 1** (Section 4). $\mathrm{ORP}_{\mathsf{graph}}(\Phi_{\mathsf{unb}}, \Psi_{\mathsf{eq}})$ is not formally verifiable.

Let $\Psi_{\mathsf{leq}}$ be a set of vector specifications, satisfied by vectors where for each dimension there is another dimension which is greater or equal. Now, $\Psi_{\mathsf{class}} := \overline{\Psi}_{\mathsf{leq}}$ is a set of vector specifications, defining vectors where a certain dimension is greater than all others or, in other words, outputs which can be interpreted as an exact class assignment. We can easily alter the proof of Theorem 1 to argue that $\mathrm{ORP}_{\mathsf{graph}}(\Phi_{\mathsf{unb}}, \Psi_{\mathsf{leq}})$ is also not formally verifiable (see Section 4). Then, Lemma 1 implies the following result for ARP of graph-classifier MPNN.

**Corollary 1.** $\mathrm{ARP}_{\mathsf{graph}}(\Phi_{\mathsf{unb}}, \Psi_{\mathsf{class}})$ is not formally verifiable.

Thus, as soon as we consider ORP or ARP of graph-classifier MPNN over parts of the input space, including graphs of unbounded, but finite size, with sufficient degree and expressive labels, it is no longer guaranteed that they are formally verifiable.

To better understand the impact of our second core contribution, we make a short note on classical NN verification. There, a common choice of specifications over vectors are conjunctions of linear inequalities $\sum_{\mathbb{I}} c_i \mathrm{x}_i \leq b$ where $c_i, b$ are rational constants and $\mathrm{x}_i$ are dimensions of a vector. Such specifications define convex sets and, thus, we call the set of all such specifications $\Psi_{\mathsf{conv}}$. Let $\Phi_{\mathsf{bound}}$ be a set of graph-node specifications, bounding the degree of valid graphs and using constraints on labels in a bounded distance to the center node which can be expressed by vector specifications as described above.[2] Now, it turns out that as soon as we bound the degree of input graphs, ORP of node-classifier MPNN with label constraints and output specifications from $\Psi_{\mathsf{conv}}$ can be verified formally.

**Theorem 2** (Section 5). $\mathrm{ORP}_{\mathsf{node}}(\Phi_{\mathsf{bound}}, \Psi_{\mathsf{conv}})$ is formally verifiable.

Again, Lemma 1 implies a similar result for ARP of node-classifier MPNN. Obviously, we have $\Psi_{\mathsf{class}} \subseteq \overline{\Psi}_{\mathsf{conv}}$.

**Corollary 2.** $\mathrm{ARP}_{\mathsf{node}}(\Phi_{\mathsf{bound}}, \Psi_{\mathsf{class}})$ is formally verifiable.

This byproduct of Theorem 2 considerably extends the set of input specifications for which ARP of node-classifier MPNN is known to be formally verifiable. In particular, the literature (see Section 1.1) gives indirect evidence that $\mathrm{ARP}_{\mathsf{node}}(\Phi_{\mathsf{neigh}}, \Psi_{\mathsf{class}})$ can be verified formally where $\Phi_{\mathsf{neigh}}$ is a set of specifications defined by a center graph and a bounded budget of allowed structural modifications as well as label alternations restricted using box constraints, which can be expressed using vector specifications of the form given above. Thus, $\Phi_{\mathsf{neigh}} \subseteq \Phi_{\mathsf{bound}}$.

The results above, in addition to some immediate implications, reveal major parts of the landscape of MPNN formal verification, depicted in Figure 1. The horizontal, resp. vertical axis represents sets

---

[2]We assume that model checking of specifications from $\Phi_{\mathsf{bound}}$ is decidable. Otherwise, verification of corresponding ORP becomes undecidable due to trivial reasons.

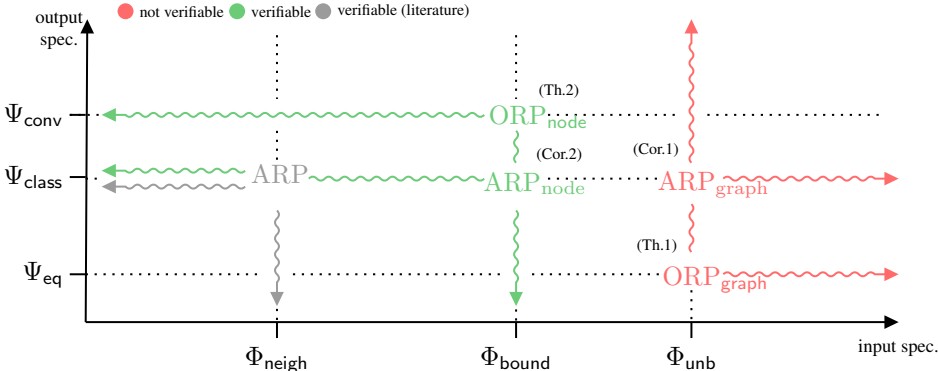

Figure 1: Overview of core results.

of input, resp. output specifications, loosely ordered by expressiveness. The three most important impressions to take from this visualisation are: first, the smaller the classes of specifications, the stronger an impossibility result becomes. Note that Theorem 1 and Corollary 1 naturally extend to more expressive classes of specifications (indicated by the red, squiggly arrows up and to the right). Second, results about the possibility to do formal verification grow in strength with the expressive power of the involved specification formalisms; Theorem 2 and Corollary 2 extend naturally to smaller classes (indicated by the green, squiggly arrows down and to the left). Third, the results presented here are not ultimately tight in the sense that there is a part of the landscape, between $\Phi_{\mathsf{bound}}$ and $\Phi_{\mathsf{unb}}$, for which the status of decidability of formal verification remains unknown.

**Remark** An interesting observation is that formal verification of ORP of node-classifier MPNN is impossible as soon as we allow for input specifications that can express properties like $\exists v \forall v' \, E(v, v')$, stating that a valid graph must contain a "master node" that is connected to all other nodes. Then the same reduction idea as seen in Section 4 can be used to show that formal verification is no longer possible.

## 4 THE IMPOSSIBILITY OF FORMALLY VERIFIYING ORP AND ARP OF GRAPH-CLASSIFIER MPNN OVER UNBOUNDED GRAPH CLASSES

The ultimate goal of this section is to show that $\mathrm{ORP}_{\mathsf{graph}}(\Phi_{\mathsf{unb}}, \Psi_{\mathsf{eq}})$ is not formally verifiable. Note that we use a weak form of $\Phi_{\mathsf{unb}}$ here. See Appendix A for details. To do so, we relate the formal verification of $\mathrm{ORP}_{\mathsf{graph}}(\Phi_{\mathsf{unb}}, \Psi_{\mathsf{eq}})$ to the following decision problem: given a graph-classifier $N$ with a single output dimension, the question is whether there is graph $\mathcal{G}$ such that $N(\mathcal{G}) = 0$. We call this problem *graph-classifier problem (GCP)*.

**Lemma 2.** If GCP is undecidable then $\mathrm{ORP}_{\mathsf{graph}}(\Phi_{\mathsf{unb}}, \Psi_{\mathsf{eq}})$ is not formally verifiable.

*Proof.* By contraposition. Suppose $\mathrm{ORP}_{\mathsf{graph}}(\Phi_{\mathsf{unb}}, \Psi_{\mathsf{eq}})$ was formally verifiable. Then there is an algorithm $A$ such that for each $(N, \mathsf{true}, \mathbf{y} = 0) \in \mathrm{ORP}_{\mathsf{graph}}(\Phi_{\mathsf{unb}}, \Psi_{\mathsf{eq}})$ we have: $A$ returns $\top$ if and only if $(N, \mathsf{true}, \mathbf{y} = 0)$ holds. But then $A$ can be used to decide GCP. $\square$

Using this lemma, in order to prove Theorem 1, it suffices to show that GCP is undecidable, which we will do in the remaining part of this section. The proof works as follows: first, we define a satisfiability problem for a logic of graphs labeled with vectors, which we call *Graph Linear Programming* (GLP) as it could be seen as an extension of ordinary linear programming on graph structures. We then prove that GLP is undecidable by a reduction from Post (1946)'s Correspondence Problem (PCP). From the form of the reduction we infer that the graph linear programs in its image are of a particular shape which can be used to define a – therefore also undecidable – fragment, called *Discrete Graph Linear Programming* (DGLP). We then show how this fragment can be reduced to GCP, thus establishing its undecidability in a way that separates the structural from the arithmetical parts in a reduction from PCP to GCP. As a side-effect, with GLP we obtain a relatively natural

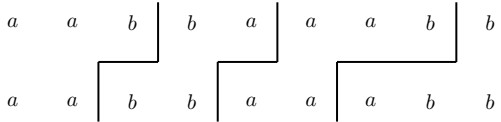

Figure 2: A solution for the PCP instance $P_0$.

undecidable problem on graphs and linear real arithmetic which may possibly serve to show further undecidability results on similar graph neural network verification problems.

## 4.1 FROM PCP TO GLP

We begin by defining the Graph Linear Programming problem GLP. Let $\mathbb{X} = \{x_1, \ldots, x_n\}$ be a set of variables. A *node condition* $\varphi$ is a formula given by the syntax

$$\varphi ::= \textstyle\sum_{i=1}^n a_i x_i + b_i(\odot x_i) \leq c \mid \varphi \wedge \varphi \mid \varphi \vee \varphi$$

where $a_j, b_j, c \in \mathbb{Q}$. Intuitively, the $x_i$ are variables for a vector of $n$ real values, constituting a graph's node label, and the operator $\odot$ describes access to the node's neighbourhood, resp. their labels.

We write $\mathrm{sub}(\varphi)$ for the set of subformulas of $\varphi$ and $\mathrm{Var}(\varphi)$ for the set of variables occurring inside $\varphi$. We use the abbreviation $t = c$ for $t \leq c \wedge -t \leq -c$.

Let $\mathcal{G} = (\mathbb{V}, \mathbb{D}, L)$ be a graph with $L : \mathbb{V} \to \mathbb{R}^n$. A node condition $\varphi$ induces a set of nodes of $\mathcal{G}$, written $[\![\varphi]\!]^{\mathcal{G}}$, and is defined inductively as follows.

$$
\begin{array}{lll}
v \in [\![\sum_{i=1}^n a_i x_i + b_i(\odot x_i) \leq c]\!]^{\mathcal{G}} & \text{iff} & \sum_{i=1}^n a_i L(v)_i + b_i(\sum_{v' \in N_v} L(v')_i) \leq c \\
v \in [\![\varphi_1 \wedge \varphi_2]\!]^{\mathcal{G}} & \text{iff} & v \in [\![\varphi_1]\!]^{\mathcal{G}} \cap [\![\varphi_2]\!]^{\mathcal{G}} \\
v \in [\![\varphi_1 \vee \varphi_2]\!]^{\mathcal{G}} & \text{iff} & v \in [\![\varphi_1]\!]^{\mathcal{G}} \cup [\![\varphi_2]\!]^{\mathcal{G}}
\end{array}
$$

If $v \in [\![\varphi]\!]^{\mathcal{G}}$ then we say that $v$ *satisfies* $\varphi$. A *graph condition* $\psi$ is a formula given by the syntax $\psi ::= \sum_{i=1}^n a_i x_i \leq c \mid \psi \wedge \psi$, where $a_i, c \in \mathbb{Q}$. The semantics of $\psi$, written $[\![\psi]\!]$, is the subclass of graphs $\mathcal{G} = (\mathbb{V}, \mathbb{D}, L)$ with $L : \mathbb{V} \to \mathbb{R}^n$ $f$such that

$$
\begin{array}{lll}
\mathcal{G} \in [\![\sum_{i=1}^n a_i x_i \leq c]\!] & \text{iff} & \sum_{i=1}^n a_i(\sum_{v \in \mathbb{V}} L(v)_i) \leq c, \\
\mathcal{G} \in [\![\psi_1 \wedge \psi_2]\!] & \text{iff} & \mathcal{G} \in [\![\psi_1]\!] \cap [\![\psi_2]\!].
\end{array}
$$

Again, if $\mathcal{G} \in [\![\psi]\!]$ then we say that $\mathcal{G}$ satisfies $\psi$.

The problem GLP is defined as follows: given a graph condition $\psi$ and a node condition $\varphi$ over the same set of variables $\mathbb{X} = \{x_1, \ldots, x_n\}$, decide whether there is a graph $\mathcal{G} = (\mathbb{V}, \mathbb{D}, L)$ with $L : \mathbb{V} \to \mathbb{R}^n$ such that $\mathcal{G}$ satisfies $\psi$ and all nodes in $\mathcal{G}$ satisfy $\varphi$. Such an $\mathcal{L} = (\psi, \varphi)$ is called a *graph linear program*, which we also abbreviate as GLP. It will also be clear from the context whether GLP denotes a particular program or the entire decision problem.

As stated above, we show that GLP is undecidable via a reduction from *Post's Correspondence Problem (PCP)*: given $P = \{(\alpha_1, \beta_1), (\alpha_2, \beta_2), \ldots, (\alpha_k, \beta_k)\} \subseteq \Sigma^* \times \Sigma^*$ for some alphabet $\Sigma$, decide whether there is a non-empty sequence of indices $i_1, i_2, \ldots, i_l$ from $\{1, \ldots, k\}$ such that $\alpha_{i_1} \alpha_{i_2} \cdots \alpha_{i_l} = \beta_{i_1} \beta_{i_2} \cdots \beta_{i_l}$. The $\alpha_i, \beta_i$ are also called *tiles*. PCP is known to be undecidable when $|\Sigma| \geq 2$, i.e. we can always assume $\Sigma = \{a, b\}$. For example, consider the solvable instance $P_0 = \{(aab, aa), (b, abb), (ba, bb)\}$. It is not hard to see that $I = 1, 3, 1, 2$ is a solution for $P_0$. Furthermore, the corresponding sequence of tiles can be visualised as shown in Figure 2. The upper word is produced by the $\alpha_i$ parts of the tiles and the lower one by the $\beta_i$. The end of one and beginning of the next tile are visualised by the vertical part of the step lines.

**Theorem 3.** GLP is undecidable.

*Proof sketch.* We sketch the proof here and give a full version in Appendix B.1. The proof is done by establishing a reduction from PCP. The overall idea is to translate each PCP instance $P$ to a GLP $\mathcal{L}_P$ with the property that $P$ is solvable if and only if $\mathcal{L}_P$ is satisfiable. Thus, the translation

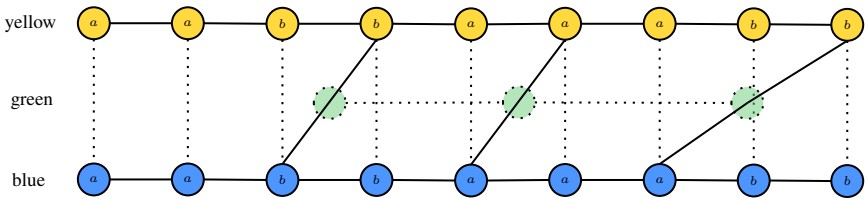

Figure 3: Encoded solution $I$ of PCP instance $P_0$.

must be such that $\mathcal{L}_P$ is only satisfiable by graphs that encode a valid solution of $P$. The encoding is depicted for the solution of $P_0$ shown in Figure 2 in form of solid lines and nodes in Figure 3. The word $w_\alpha = \alpha_1\alpha_3\alpha_1\alpha_2$ is represented by the chain of yellow nodes from left to right in such way that there is a node for each symbol $w_i$ of $w_\alpha$. If $w_i = a$ then $x_a = 1$ and $x_b = 0$ of the corresponding node and vice-versa if $w_i = b$. Analogously, $\beta_1\beta_3\beta_1\beta_2$ is represented by the blue chain. The borders between two tiles are represented as edges between the yellow and blue nodes corresponding to the starting positions of a tile. The encoding as a graph uses additional auxiliary nodes, edges and label dimensions, in order to ensure that the labels along the yellow and blue nodes indeed constitute a valid PCP solution, i.e. the sequences of their letter labels are the same, and they are built from corresponding tiles in the same order. In Figure 3, these auxiliary nodes and edges are indicated by the dashed parts. □

GLP seems to be too expressive in all generality for a reduction to GCP, at least it does not seem (easily) possible to mimic arbitrary disjunctions in an MPNN. However, the node conditions $\varphi$ resulting from the reduction from PCP to GLP are always of a very specific form: $\varphi = \varphi' \wedge \varphi_{\mathrm{discr}}$ where $\varphi_{\mathrm{discr}} = \bigwedge_{i \in I} \bigvee_{m \in \mathbb{M}} x_i = m$ with $\mathbb{M} \subset \mathbb{N}$ enforces dimensions $x_i$ to be discrete and $\varphi'$ has the following property. Let $\mathbb{X}$ be the set of dimensions not discretized by $\varphi_{\mathrm{discr}}$. For each $\varphi_1 \vee \varphi_2 \in \mathrm{sub}(\varphi')$ it is the case that $\mathrm{Var}(\varphi_1) \cap \mathbb{X} = \emptyset$ or $\mathrm{Var}(\varphi_2) \cap \mathbb{X} = \emptyset$. In other words, in each disjunction in $\varphi'$ at most one disjunct contains non-discretised dimensions. We call this fragment of GLP *Discrete Graph Linear Programming* (DGLP). The observation that the reduction function from PCP constructs graph linear programs which fall into DGLP (see Appendix B.1 for details) immediately gives us the following result.

**Corollary 3.** DGLP is undecidable.

## 4.2 FROM DGLP TO GCP

**Theorem 4.** GCP is undecidable.

*Proof.* By a reduction from DGLP. Given a DGLP $\mathcal{L} = (\varphi, \psi)$ we construct an MPNN $N_\mathcal{L}$ that gives a specific output, namely 0, if and only if its input graph $\mathcal{G}$ satisfies $\mathcal{L}$ and therefore is a witness for $\mathcal{L} \in$ GLP. Let $m, n \in \mathbb{R}$ with $m \leq n$ and $\mathbb{M} = \{i_1, i_2, \ldots, i_k\} \subseteq \mathbb{N}$ such that $i_j \leq i_{j+1}$ for all $j \in \{1, \ldots, k-1\}$. We use the auxiliary gadget $\langle x \in [m; n] \rangle := \mathrm{re}(\mathrm{re}(x - n) - \mathrm{re}(x - (n+1)) + \mathrm{re}(m - x) + \mathrm{re}((m-1) - x))$ to define the gadgets

$$\langle x \leq m \rangle := \mathrm{re}(\mathrm{re}(x - m) - \mathrm{re}(x - (m+1))) \text{ and}$$

$$\langle x \in \mathbb{M} \rangle := \mathrm{re}\left( \langle x \in [i_1; i_k] \rangle + \sum_{j=1}^{k-1} \mathrm{re}\left( \tfrac{(i_{j+1} - i_j)}{2} - (\mathrm{re}(x - \tfrac{i_j + i_{j+1}}{2}) + \mathrm{re}(\tfrac{i_j + i_{j+1}}{2} - x)) \right) \right).$$

Each of the gadgets above fulfils specific properties which can be inferred from their functional forms without much effort: let $r \in \mathbb{R}$. Then, $\langle r \leq m \rangle = 0$ if and only if $r \leq m$, and $\langle r \in \mathbb{M} \rangle = 0$ if and only if $r \in \mathbb{M}$. Furthermore, both gadgets are positive and $\langle x \leq m \rangle$ is upwards bounded for all $m$ by 1 with the property that $|r - m| \geq 1$ implies $\langle r \leq m \rangle = 1$. We give a formal proof in Appendix B.2. We use $\langle x = m \rangle$ as an abbreviation for $\langle -x \leq -m \rangle + \langle x \leq m \rangle$.

The input size of $N_\mathcal{L}$ equals the amount of variables occurring in $\varphi$ and $\psi$. $N_\mathcal{L}$ has one layer with two output dimensions $y^1_{\mathrm{discr}}$ and $y^1_{\mathrm{cond}}$ and the readout layer has a single output dimension $y^r$. The subformula $\varphi_{\mathrm{discr}} = \bigwedge_{i \in \mathbb{I}} \bigvee_{m \in \mathbb{M}_i} x_i = m$ is represented by $y^1_{\mathrm{discr}} = \sum_{i \in I} \langle x_i \in \mathbb{M}_i \rangle$ and then checked using $\langle y^1_{\mathrm{discr}} = 0 \rangle$ in the readout layer.

The remaining part of $\varphi$ is represented in output dimension $y^1_{\mathrm{cond}}$ in the following way. Obviously, an atomic $\leq$-formula is represented using a $\langle x \leq m \rangle$ gadget. A conjunction $\varphi_1 \wedge \varphi_2$ is represented

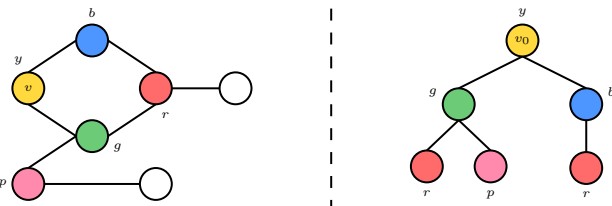

Figure 4: Tree-model property of a two-layered MPNN.

by a sum of two gadgets $f_1 + f_2$ where $f_i$ represents $\varphi_i$. For this to work, we need the properties that all used gadgets are positive and that their output is 0 when satisfied.

To represent a disjunction $\varphi_1 \vee \varphi_2$ where $f_1$ and $f_2$ are the gadgets representing $\varphi_1$ resp. $\varphi_2$, we need the fact that $\mathcal{L}$ is a DGLP. W.l.o.g. suppose that $\varphi_1$ only contains discrete variables and that $\varphi_{\text{discr}}$ is satisfied. Then we get: if $\varphi_1$ is not satisfied then the output of $f_1$ must be greater or equal to 1. The reason for this is the following. If the property of some $\langle \text{x} \leq m \rangle$-gadget is not satisfied its output must be 1, still under the assumption that its input includes discrete variables only. Furthermore, as $\langle \text{x} \leq m \rangle$ is positive and upwards bounded, the value of $f_2$ must be bounded by some value $k \in \mathbb{R}^{>0}$. Therefore, we can represent the disjunction using $\text{re}(f_2 - k \, \text{re}(1 - f_1))$. Note that this advanced gadget is also positive and upwards bounded. Again, the value of $\text{y}^1_{\text{cond}}$ is checked in the readout layer using $\langle \text{y}^1_{\text{cond}} = 0 \rangle$. The graph condition $\psi$ is represented using a sum of $\langle \text{x} \leq m \rangle$ gadgets.

Thus, we can effectively translate a DGLP $\mathcal{L}$ into an MPNN $N_{\mathcal{L}}$ such that there is a graph $\mathcal{G}$ with $N_{\mathcal{L}}(\mathcal{G}) = 0$ if and only if $\mathcal{G}$ satisfies $\mathcal{L}$, i.e. $\mathcal{L} \in \text{DGLP}$. This transfers the undecidability from DGLP to GCP. □

**Proof of Theorem 1.** The statement is an immediate consequence of the results of Theorem 4 and Lemma 2. □

The proof for Corollary 1 follows the exact same line of arguments, but we consider the following decision problem: given a graph-classifier $N$ with two output dimension, the question is whether there is graph $\mathcal{G}$ such that $(N(\mathcal{G})_1 \leq N(\mathcal{G})_2) \wedge (N(\mathcal{G})_1 \leq N(\mathcal{G})_2)$. We call this $\text{GCP}_<$. Obviously, the statement of Lemma 2 also holds for $\text{GCP}_<$ and $\text{ORP}_{\text{graph}}(\Phi_{\text{bound}}, \Psi_{\text{leq}})$. Proving that $\text{GCP}_<$ is undecidable is also done via reduction from DGLP with only minimal modifications of MPNN $N_{\mathcal{L}}$ constructed in the proof of Theorem 4: we add a second output dimension to $N_{\mathcal{L}}$ which constantly outputs 0. The correctness of the reduction follows immediately.

## 5 THE POSSIBILITY OF FORMALLY VERIFIYING ORP AND ARP OF NODE-CLASSIFIER MPNN OVER DEGREE BOUNDED GRAPH CLASSES

In order to prove Theorem 2, we argue that there is a naive algorithm verifying $\text{ORP}_{\text{node}}(\Phi_{\text{bound}}, \Psi_{\text{conv}})$ formally. Consider a node-classifier $N$ with $k$ layers and consider some graph $\mathcal{G}$ with specified node $v$ such that $N(\mathcal{G}, v) = \boldsymbol{y}$. The crucial insight is that there is a tree $\mathcal{B}$ of finite depth $k$ and with root $v_0$ such that $N(\mathcal{B}, v_0) = \boldsymbol{y}$. The intuitive reason for this is that $N$ can update the label of node $v$ using information from neighbours of $v$ of distance at most $k$. For example, assume that $k = 2$ and $\mathcal{G}, v$ are given as shown on the left side of Figure 4 where the information of a node, given by its label, is depicted using different colours $y$ (yellow), $b$ (blue), $r$ (red), $g$ (green) and $p$ (pink). As $N$ only includes two layers, information from the unfilled (white) nodes are not relevant for the computation of $N(\mathcal{G}, v)$ as their distance to $v$ is greater than 2. Take the tree $\mathcal{B}$ on the right side of Figure 4. We get that $N(\mathcal{G}, v) = N(\mathcal{B}, v_0)$.

**Proof of Theorem 2.** First, we observe the tree-model property for node-classifier MPNN over graphs of bounded degree: let $(N, \varphi, \psi) \in \text{ORP}_{\text{node}}(\Phi_{\text{bound}}, \Psi_{\text{conv}})$ where $N$ has $k'$ layers and $\varphi$ bounds valid graphs to degree $d \in \mathbb{N}$ and constraints nodes in the $k''$-neighbourhood of the center node. We have that $(N, \varphi, \psi)$ holds if and only if there is a $d$-tree $\mathcal{B}$ of depth $k = \max(k', k'')$ with root $v_0$ such that $(\mathcal{B}, v_0) \models \varphi$ and $N(\mathcal{B}, v_0) \models \psi$. We prove this property in Appendix B.3.

We fix the ORP $(N, \varphi, \psi)$ as specified above and assume that $\mathrm{comb}_i$ as well as the readout function *read* of $N$ are given by the NN $N_1, \ldots, N_{k'}, N_r$ where $N_1$ has input dimension $2 \cdot m$ and output dimension $n$. Furthermore, assume that $\varphi$ bounds valid graphs to degree $d \in \mathbb{N}$. For each unlabeled tree $\mathcal{B} = (\mathbb{V}, \mathbb{D}, v_0)$ with $\mathbb{V} = \{v_0, \ldots, v_l\}$ of degree at most $d$ and depth $k$, of which there are only finitely many, the verification algorithm $A$ works as follows.

By definition, the MPNN $N$ applied to $\mathcal{B}$ computes $N_1(\boldsymbol{x}_v, \sum_{v' \in N_v} \boldsymbol{x}_{v'})$ as the new label for each $v \in \mathbb{V}$ after layer $l_1$. However, as the structure of $\mathcal{B}$ is fixed at this point we know the neighbourhood for each node $v$. Therefore, $A$ constructs NN $\mathcal{N}_1$ with input dimension $l \cdot m$ and output dimension $l \cdot n_1$ given by $(N_1(\mathrm{id}(\mathbf{x}_{v_0}), \mathrm{id}(\sum_{v' \in N_{v_0}}(\mathbf{x}_{v'}))), \ldots, N_1(\mathrm{id}(\mathbf{x}_{v_l}), \mathrm{id}(\sum_{v' \in N_{v_l}}(\mathbf{x}_{v'}))))$, representing the whole computation of layer $l_1$, where $\mathrm{id}(x) := \mathrm{re}(\mathrm{re}(x) - \mathrm{re}(-x))$ is a simple gadget computing the identity. In the same way $A$ transforms the computation of layer $l_i$, $i \geq 2$, into a network $\mathcal{N}_i$ using the output of $\mathcal{N}_{i-1}$ as inputs. Then, $A$ combines $\mathcal{N}_l$ and $N_r$, by connecting the output dimensions of $\mathcal{N}_l$ corresponding to node $v_0$ to the input dimensions of $N_r$, creating an NN $\mathcal{N}$ representing the computation of $N$ over graphs of structure $\mathcal{B}$ for arbitrary labeling functions $L$.

This construction reduces the question of whether $(N, \varphi, \psi)$ holds to the following question: are there labels for $v_0, \ldots, v_l$, the input of $\mathcal{N}$, satisfying the constraints given by $\varphi$ such that the output of $\mathcal{N}$ satisfies $\psi$. As the label constraints of $\varphi$ and the specification $\psi$ are defined by conjunctions of linear inequalities this is in fact an instance of the output reachability problem for NN, which is known to be decidable, as shown by Katz et al. (2017) or Sälzer & Lange (2021). Therefore, $A$ incorporates a verification procedure for ORP of NN and returns $\top$ if the instance is positive and otherwise considers the next unlabeled $d$-tree of depth $k$. If none has been found then it returns $\bot$. The soundness and completeness of $A$ follows from the tree-model property, the exhaustive loop over all candidate trees and use of the verification procedure for output reachability of NN.

## 6 SUMMARY AND APPLICABILITY OF RESULTS

This work presents two major results: we proved that formal verification of ORP and ARP of graph-classifier MPNN is not possible as soon as we consider parts of the input space, containing graphs of unbounded, but finite size, non-trivial degree and sufficiently expressive labels. We also showed that formal verification of ORP and ARP of node-classifier MPNN is possible, as soon as the degree of the considered input graphs is bounded. These results can serve as a basis for further research on formal verification of GNN but their extendability depends on several parameters.

**Dependency on the GNN model.** We restricted our investigations to GNN from the MPNN model, which is a blueprint for spatial-based GNN (Wu et al. (2021)). However, the MPNN model does not directly specify how the aggregation, combination and readout functions are represented. Motivated by common choices, we restricted our considerations to MPNN where the aggregation functions compute a simple sum of their inputs and the combination and readout functions are represented by NN with ReLU activation only. Theorem 1 and Corollary 1 only extend to GNN models that are *at least* as expressive as the ones considered here. For some minor changes to our setting, like considering NN with other piecewise-linear activation functions, it is easily argued that both results still hold. However, as soon as we leave the MPNN or spatial-based model the question of formal verifiability opens anew. Bridging results about the expressiveness of GNN from different models, for example spatial- vs. spectral-based, is ongoing research like done by Balcilar et al. (2021), and it remains to be seen which future findings on expressiveness can be used to directly transfer the negative results about the impossibility of formal verification obtained here. Analogously, Theorem 2 and Corollary 2 only extend to GNN that are *at most* as expressive as the ones considered here. It is not possible, for example, to directly translate these results to models like DropGNN (Papp et al. (2021)), which are shown to be more expressive than MPNN. Hence, this also remains to be investigated in the future.

**Dependency on the specifications.** Obviously, the results presented here are highly dependent on the choice of input as well as output specifications (see Appendix A for details). We refer to future work for establishing further (im-)possibility results for formal verification of ORP and ARP of GNN, with the ultimate goal of finding tight bounds.

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

# A    DETAILS ON IMPORTANT SETS OF SPECIFICATIONS

To prove the results stated in the main part of the paper, we need to work with a formally defined syntax for each kind of specification considered here. However, it should be clear that the results presented in Section 3 do not depend on the exact syntactic form, but on the expressibility of the considered kind of specifications.

**Vector Specifications $\Psi_{\text{conv}}$.** Motivated by common choices in formal verification of classical NN, we often use the following form: a *vector specification* $\varphi$ for a given set of variables $\mathbb{X}$ is defined by the grammar

$$\varphi ::= \varphi \wedge \varphi \mid t \leq b , \quad t ::= c \cdot \mathrm{x} \mid t + t$$

where $b, c \in \mathbb{Q}$ and $\mathrm{x} \in \mathbb{X}$ is a variable. A vector specification $\varphi$ with occurring variables $\mathrm{x}_0, \ldots, \mathrm{x}_{n-1}$ is satisfied by $\boldsymbol{x} = (r_0, \ldots, r_{n-1}) \in \mathbb{R}^n$ if each inequality in $\varphi$ is satisfied in real arithmetic with each $\mathrm{x}_i$ set to $r_i$. We denote the set of all such specifications by $\Psi_{\text{conv}}$. A vector specification that also includes $\vee$ and $<$ operators is called *extended*. Extended vector specifications are not included in $\Psi_{\text{conv}}$.

**Graph specifications from $\Phi_{\text{unb}}$.** In the arguments of Section 4 and Appendix B.1 we refer to any set of graph specifications which contains a specification $\varphi$ satisfiable by finite graphs of arbitrary size and degree as well as arbitrary labels, as $\Phi_{\text{unb}}$, for instance a specification like $\varphi = \text{true}$. However, as indicated in Section 3, this weak form of $\Phi_{\text{unb}}$ is not necessary. The arguments for Theorem 1, given in Section 4 and Appendix B.1, are also valid if $\Phi_{\text{unb}}$ contains at least one specification $\varphi$, satisfiable by graphs of arbitrary size, degree 4 (indicated by Figure 3) and labels expressive enough to represent those used in PCP-structures (Appendix B.1), mainly labels using positive integer values. For example, $\varphi(\mathcal{G}) = \deg(\mathcal{G}) \leq 4 \wedge \forall v \in \mathbb{V}. \psi_{\text{PCP}}(v)$, where $\psi_{\text{PCP}}$ is an extended vector specification, checking if the label of a node is valid for a PCP-structure. However, the exact conditions are highly dependent on the construction used in the reduction from PCP to GLP and can easily be optimised. But the aim of this work is to show that there are fundamental limits in formal verification of ORP (and ARP), and optimising the undecidability results presented in Section 4 in this way would only obscure the understanding of such limits. Thus, we use the above described weaker $\Phi_{\text{unb}}$ in the formal parts, leading to uncluttered arguments and proofs.

**Graph-node specifications from $\Phi_{\text{bound}}$.** First, we define for each $d, k \in \mathbb{N}$ a set $\Phi_{\text{bound}}^{d,k}$ of graph-node specifications. $\Phi_{\text{bound}}^{d,k}$ is the set graph-node specifications $\varphi$ bounding the degree of satisfying graphs to $d$ and constraining only nodes in the $k$-neighbourhood of the center node using vector specifications, for instance $\varphi(\mathcal{G}, v) = \deg(\mathcal{G}) \leq 4 \wedge \forall v' \in \text{Neigh}_k(v). \psi(v')$ where $\text{Neigh}_k$ includes all nodes of distance up to $k$ of $v$ and $\psi$ is a vector specification. Then, $\Phi_{\text{bound}} = \bigcup_{d,k \in \mathbb{N}} \Phi_{\text{bound}}^{d,k}$.

**Graph or graph-node specifications from $\Phi_{\text{neigh}}$.** We consider $\Phi_{\text{neigh}}$ as a set of graph specifications or a set of graph-node specifications. $\Phi_{\text{neigh}}$ consists of graph or graph-node specifications $\varphi$, given by some fully-defined center graph $\mathcal{G}$ (or pair $(\mathcal{G}, v)$) and a finite modification-budget $B$. A finite modification budget specifies a bounded number of structural modifications, namely inserting or deleting nodes and edges, as well as allowed label modifications of nodes in $\mathcal{G}$, bounded by vector specifications, for instance $\varphi = (\mathcal{G}, B)$ or $\varphi = ((\mathcal{G}, v), B)$. Then, a graph or graph-node pair satisfies $\varphi$ if it can be generated from $\mathcal{G}$ respectively $(\mathcal{G}, v)$ respecting the bounded budget $B$.

**Vector specifications $\Psi_{\text{eq}}$.** The set $\Psi_{\text{eq}}$ consists of vector specifications of the form $\mathrm{x}_i = b$, thus, vector specifications expressing that a single dimension is equal to some fixed, rational value.

**Extended vector specifications $\Psi_{\text{leq}}, \Psi_{\text{class}}$.** The $\Psi_{\text{leq}}$ consists of extended vector specifications of the form $\bigwedge_{i \in \mathbb{I}} \bigvee_{j \in \mathbb{I} \setminus \{i\}} \mathrm{x}_i \leq \mathrm{x}_j$. Analogously, $\Psi_{\text{class}}$ consists of extended vector specifications of the form $\bigvee_{i \in \mathbb{I}} \bigwedge_{j \in \mathbb{I} \setminus \{i\}} \mathrm{x}_i > \mathrm{x}_j$. Note that for the argument of Corollary 1 it is sufficient that $(\mathrm{x}_1 \leq \mathrm{x}_2) \wedge (\mathrm{x}_2 \leq \mathrm{x}_1)$ is included in $\Psi_{\text{leq}}$.

# B  PROOF DETAILS

## B.1  PROVING THAT GLP AND DGLP ARE UNDECIDABLE

We use the following abbreviations for GLP. For a set $\mathbb{C}$ of colours we define $\text{colour}(\mathbb{C}) = \bigwedge_{\mathbb{C}} (\mathrm{x}_c = 0) \vee (\mathrm{x}_c = 1)$ and $\text{exactly\_one}(\mathbb{C}) = \bigwedge_{\mathbb{C}} (c \rightarrow (\bigwedge_{c' \neq c} \neg c')) \wedge (\neg c \rightarrow \bigvee_{c' \neq c} c')$ where $c := (\mathrm{x}_c = 1)$, $\neg c := (\mathrm{x}_c = 0)$, $c \rightarrow \varphi := (\mathrm{x}_c = 0) \vee \varphi$ and $\neg c \rightarrow \varphi := (\mathrm{x}_c = 1) \vee \varphi$. We use $\rightarrow$ as having a weaker precedence than all other GLP operators. To keep the notation clear we denote – if unambiguous – some variable $\mathrm{x}_i$ in node and graph conditions by its index $i$.

Let $\mathcal{G} = (\mathbb{V}, \mathbb{D}, L)$ be a graph. For some node set $\mathbb{V}' \subseteq \mathbb{V}$ and node $v$ we define $\mathrm{Neigh}_v(\mathbb{V}') = \mathrm{Neigh}(v) \cap \mathbb{V}'$. We call a subset of nodes $\mathbb{V}' = \{v_1, \ldots, v_k\} \subseteq \mathbb{V}$, $k \geq 2$, a *chain* if $\mathrm{Neigh}_{v_1}(\mathbb{V}') = \{v_2\}$, $\mathrm{Neigh}_{v_i}(\mathbb{V}') = \{v_{i-1}, v_{i+1}\}$ for $2 \leq i \leq k-1$ and $\mathrm{Neigh}_{v_k}(\mathbb{V}') = \{v_{k-1}\}$. We call $v_1$ *start*, $v_i$ a *middle* node and $v_k$ *end* of $\mathbb{V}'$ and assume throughout the following arguments that index 1 denotes the start and the maximal index $k$ denotes the end of a chain. Let $\mathbb{V}_1 = \{v_1, \ldots, v_k\}$ and $\mathbb{V}_2 = \{u_1, \ldots, u_k\}$ be subsets of $\mathbb{V}$ and both be chains. We say that $\mathbb{V}_1 \cup \mathbb{V}_2$ is a *ladder* if for all $v_i, u_i$ we have $\mathrm{Neigh}_{v_i}(\mathbb{V}_1 \cup \mathbb{V}_2) = \mathrm{Neigh}_{v_i}(\mathbb{V}_1) \cup \{u_i\}$ and $\mathrm{Neigh}_{u_i}(\mathbb{V}_1 \cup \mathbb{V}_2) = \mathrm{Neigh}_{u_i}(\mathbb{V}_2) \cup \{v_i\}$.

First, we show that DGLP can recognise graphs $\mathcal{G}$ that consist of exactly one ladder and one additional chain. If this is the case we call $\mathcal{G}$ a *chain-ladder*. Let $\mathbb{C}_3 = \{c_1, c_2, c_3\}$, $\mathbb{T} = \{(c,s), (c,m), (c,e) \mid c \in \mathbb{C}_3\}$ be sets of of symbols we call *colours*. Let $(\varphi_{\mathrm{CL}}, \psi_{\mathrm{CL}})$ be the following DGLP over variables $\mathrm{Var}_{\mathrm{CL}} = \{\mathrm{x}_c \mid c \in \mathbb{C}_3 \cup \mathbb{T}\} \cup \{\mathrm{x}_{c,\mathrm{id}}, \mathrm{x}_{c,e,\mathrm{id}} \mid c \in \mathbb{C}_3\}$:

$$\varphi_{\mathrm{CL}} := \varphi_{\mathrm{cond}} \wedge \mathrm{colour}(\mathbb{C}_3 \cup \mathbb{T}) \wedge \bigwedge\nolimits_{\mathbb{M} \in \{\mathbb{C}_3, \mathbb{T}\}} \mathrm{exactly\_one}(\mathbb{M})$$

$$\varphi_{\mathrm{cond}} := \bigwedge\nolimits_{\mathbb{C}_3} (\neg c_i \rightarrow \bigwedge\nolimits_{\mathbb{T}} \neg(c_i, t) \wedge (c_i, \mathrm{id}) = 0 \wedge (c_i, e, \mathrm{id}) = 0) \wedge (c_i \rightarrow \odot c_i = 1 \vee \odot c_i = 2)$$

$$\wedge ((c_i, s) \rightarrow \odot c_i = 1 \wedge (c_i, \mathrm{id}) = 1 \wedge \odot(c_i, \mathrm{id}) = 2 \wedge (c_i, e, \mathrm{id} = 0))$$

$$\wedge ((c_i, m) \rightarrow \odot c_i = 2 \wedge 2(c_i, \mathrm{id}) = \odot(c_i, \mathrm{id}) \wedge (c_i, e, \mathrm{id} = 0))$$

$$\wedge ((c_i, e) \rightarrow \odot c_i = 1 \wedge (c_i, \mathrm{id}) \leq \odot(c_i, \mathrm{id}) - 1 \wedge (c_i, e, \mathrm{id}) = (c_i, \mathrm{id}))$$

$$\wedge (c_1 \rightarrow \odot c_2 = 1 \wedge (c_1, \mathrm{id}) = \odot(c_2, \mathrm{id})) \wedge (c_2 \rightarrow \odot c_1 = 1 \wedge (c_2, \mathrm{id}) = \odot(c_1, \mathrm{id}))$$

$$\psi_{\mathrm{CL}} := \bigwedge\nolimits_{\mathbb{C}_3} (c_i, s) = 1 \wedge (c_i, e) = 1 \wedge c_i = (c_i, e, \mathrm{id})$$

**Lemma 3.** If $\mathcal{G} = (\mathbb{V}, \mathbb{D}, L)$ satisfies $(\varphi_{\mathrm{CL}}, \psi_{\mathrm{CL}})$ then $\mathcal{G}$ is a chain-ladder and if $\mathcal{G}' = (\mathbb{V}', \mathbb{D}')$ is an unlabeled chain-ladder then there is $L'$ such that $\mathcal{G} = (\mathbb{V}', \mathbb{D}', L')$ satisfies $(\varphi_{\mathrm{CL}}, \psi_{\mathrm{CL}})$.

*Proof.* Assume that $\mathcal{G}$ satisfies $(\varphi_{\mathrm{CL}}, \psi_{\mathrm{CL}})$. By definition, it follows that all nodes $v \in \mathbb{V}$ satisfy $\varphi_{\mathrm{CL}}$ and $\mathcal{G}$ satisfies $\psi_{\mathrm{CL}}$.

Let $v \in \mathbb{V}$ be a node. Due to $\bigwedge_{\mathbb{M} \in \{\mathbb{C}_3, \mathbb{T}\}} \mathrm{exactly\_one}(\mathbb{M}) \wedge \mathrm{colour}(\mathbb{C}_3 \cup \mathbb{T})$ we have that $v$ has exactly one colour $c_1$, $c_2$ or $c_3$ and exactly one from $\mathbb{T}$. Furthermore, the subformula $\bigwedge_{\mathbb{C}_3} (\neg c_i \rightarrow \bigwedge_{\mathbb{T}} \neg(c_i, t) \wedge \cdots$ implies that there is $i \in \{1, 2, 3\}$ such that $v$ is of colour $c_i$ and $(c_i, t)$ for some $t$.

We divide $\mathbb{V}$ into three sets $\mathbb{V}_1$, $\mathbb{V}_2$ and $\mathbb{V}_3$ such that $v \in \mathbb{V}_i$ if and only if $v$ is of colour $c_i$ and argue that each $\mathbb{V}_i$ is a chain. Note that the $\mathbb{V}_i$ are disjoint sets. Let $v \in \mathbb{V}_i$. The subformula $(c_i \rightarrow \odot c_i = 1 \vee \odot c_i = 2)$ implies that $v$ has 1 or two neighbours from $\mathbb{V}_i$. From the argument above, we know that $v$ must be of exactly one colour $(c_i, s)$, $(c_i, m)$ or $(c_i, e)$. The $\rightarrow$ subformulas in $\varphi_{\mathrm{cond}}$ regarding these three colours imply: if $v$ is of colour $(c_i, s)$ or $(c_i, e)$ it must have exactly one neighbour from $\mathbb{V}_i$ and if $v$ is of colour $(c_i, m)$ it must have exactly two neigbours from $\mathbb{V}_i$. The graph condition $\psi_{\mathrm{CL}}$ implies that there is exactly one node with colour $(c_i, s)$ and one with colour $(c_i, e)$. In combination, we have that there is a start $v_s$ and end $v_e$ in $\mathbb{V}_i$ both having one neighbour in $\mathbb{V}_i$ and all middle nodes $v_m$ having two.

Next, consider the $(c_i, \mathrm{id})$ and $(c_i, e, \mathrm{id})$ label dimensions. We call $(c_i, \mathrm{id})$ the *id* of a node with colour $c_i$. The subformula $(\neg c_i \rightarrow \cdots \wedge (c_i, \mathrm{id}) = 0 \wedge (c_i, e, \mathrm{id}) = 0) \wedge \cdots)$ implies that if a node is not of colour $c_i$ then the corresponding dimensions must be 0 and $((c_i, s) \rightarrow \cdots \wedge (c_i, e, \mathrm{id} = 0))$ and $((c_i, m) \rightarrow \cdots \wedge (c_i, e, \mathrm{id} = 0))$ imply that if it is not an end node then $(c_i, e, \mathrm{id})$ is 0 as well. Next, we see in the subformula $((c_i, s) \rightarrow \cdots \wedge (c_i, \mathrm{id}) = 1 \wedge \odot(c_i, \mathrm{id}) = 2 \wedge \cdots)$ that $v_s$ has id 1 and its neighbour has id 2. This implies that the only neighbour of $v_s$ is not itself. The same holds for $v_e$ due to the subformula $((c_i, e) \rightarrow \cdots \wedge (c_i, \mathrm{id}) \leq \odot(c_i, \mathrm{id}) - 1)$. Furthermore, the subformula $((c_i, e) \rightarrow \cdots \wedge (c_i, e, \mathrm{id}) = (c_i, \mathrm{id}))$ implies that the id of $v_e$ is stored in $(c_i, e, \mathrm{id})$. This is used in the graph condition subformula $c_i = (c_i, e, \mathrm{id})$ to ensure that the amount of nodes in $\mathbb{V}_i$ is equal to the id of $v_e$. We make a case distinction: if $v_e$ is the neighbour of $v_s$ then the id of $v_e$ is 2 and, thus, $\mathbb{V}_1 = \{v_s, v_e\}$ which obviously is a chain. If $v_e$ is not the neighbour of $v_s$ then it must be some $v_m$. The subformula $((c_i, m) \rightarrow \cdots \wedge 2(c_i, \mathrm{id}) = \odot(c_i, \mathrm{id}) \wedge \cdots)$ implies that $v_m$ is not its own neighbour and that the other neighbour $v'_m$ must have id 3. Now, if $v'_m = v_e$ then we can make the same argument as in the other case. If not then we get that $v'_m$ must have a neighbour $v''_m \neq v'_m$. The node $v''_m$ must have id 4 and, thus, it did not occur earlier on the chain. As $\mathbb{V}_i$ is finite, this sequence must eventually reach $v_e$ and we get that $\mathbb{V}_i$ must be a chain.

So far, we argued that $\mathbb{V} = \mathbb{V}_1 \cup \mathbb{V}_2 \cup \mathbb{V}_3$ with $\mathbb{V}_i$ disjoint and chains. It is left to argue that $\mathbb{V}_1 \cup \mathbb{V}_2$ forms a ladder. From our previous arguments we know that the nodes of $\mathbb{V}_i$ have incrementing

ids from $v_s$ to $v_e$ starting with 1. Therefore, the ladder property is ensured by the subformulas $(c_1 \rightarrow \odot c_2 = 1 \wedge (c_1, \mathrm{id}) = \odot(c_2, \mathrm{id}))$ and $(c_2 \rightarrow \odot c_1 = 1 \wedge (c_2, \mathrm{id}) = \odot(c_1, \mathrm{id}))$.

The other statement of the lemma, namely that there is a labeling function $L'$ for $\mathcal{G}'$ such that $(\varphi_{\mathrm{CL}}, \psi_{\mathrm{CL}})$ is satisfied, is a straightforward construction of $L'$ following the arguments above. $\quad\square$

Let $\mathcal{G}$ be a chain-ladder with ladder $\mathbb{V}_1 \cup \mathbb{V}_2 = \{v_1, \ldots, v_k\} \cup \{u_1, \ldots, u_k\}$ and chain $\mathbb{V}_3 = \{w_1, \ldots, w_l\}$. We call $\mathcal{G}$ a *PCP-structure* if for all $w_i$ we have $\mathrm{Neigh}_{w_i}(\mathbb{V}_1 \cup \mathbb{V}_2 \cup \mathbb{V}_3) = \mathrm{Neigh}_{w_i}(\mathbb{V}_3) \cup \{v_{h_i}, u_{j_i}\}$ for some $h_i, j_i \in \{1, \ldots, k\}$ such that for all $2 \le i \le l-1$ we have that $h_{i-1} < h_i < h_{i+1}$ and $j_{i-1} < j_i < j_{i+1}$. Intuitively, this property ensures that connections from chain $\mathbb{V}_3$ to $\mathbb{V}_1$ or $\mathbb{V}_2$ do not intersect.

We show that there is a DGLP that recognizes PCP-structures. Let $L, M, R$ be colours, called *directions*. We define $L + 1 := M, M + 1 := R, R + 1 := L$ and $direction - 1$ symmetrically. Let $\mathbb{C}_3$ be as above and $\mathbb{F} = \{(c, d) \mid c \in \mathbb{C}_3, d \in \{L, M, R\}\}$. The DGLP $(\varphi_{\mathrm{PS}}, \psi_{\mathrm{PS}})$ over the variables $\mathrm{Var}_{\mathrm{PS}} = \mathrm{Var}_{\mathrm{CL}} \cup \{\mathrm{x}_c \mid c \in \mathbb{F}\} \cup \{\mathrm{x}_{d, c_i, \mathrm{id}} \mid d \in \{L, M, R\}, c_i \in \{c_1, c_2\}\}$ is defined as follows:

$$\varphi_{\mathrm{PS}} := \varphi_{\mathrm{cond}} \wedge \mathrm{colour}(\mathbb{F}) \wedge \mathrm{exactly\_one}(\mathbb{F}) \wedge \varphi_{\mathrm{CL}}$$

$$\begin{aligned}
\varphi_{\mathrm{cond}} := & \left(\bigwedge_{\mathbb{C}_3} \neg c_i \rightarrow \bigwedge_{\mathbb{F}} \neg(c_i, d)\right) \\
& \wedge \bigwedge_{\mathbb{C}_3} ((c_i, s) \rightarrow (c_i, L) \wedge \odot(c_i, M) = 1) \\
& \quad\quad \wedge ((c_i, m) \rightarrow \bigvee_{\mathbb{F}} (c_i, d) \wedge \bigwedge_{d' \neq d} \odot(c_i, d') = 1) \\
& \wedge \left(\bigwedge_{\{c_1, c_2\}} c_i \rightarrow \odot c_3 \le 1\right) \wedge (c_3 \rightarrow \odot c_1 = 1 \wedge \odot c_2 = 1) \\
& \wedge \bigwedge_{\mathbb{F}} (\neg(c_3, d) \rightarrow \bigwedge_{\{c_1, c_2\}} (d, c_i, \mathrm{id}) = 0) \wedge ((c_3, d) \rightarrow \bigwedge_{\{c_1, c_2\}} \odot(c_i, \mathrm{id}) = (d, c_i, \mathrm{id})) \\
& \wedge \bigwedge_{\mathbb{F}} ((c_3, d) \rightarrow \bigwedge_{\{c_1, c_2\}} \odot(d-1, c_i, \mathrm{id}) \le (d, c_i, \mathrm{id}) \wedge (d, c_i, \mathrm{id}) \le \odot(d+1, c_i, \mathrm{id}))
\end{aligned}$$

$$\psi_{\mathrm{PS}} := \psi_{\mathrm{CL}}$$

**Lemma 4.** If $\mathcal{G} = (\mathbb{V}, \mathbb{D}, L)$ satisfies $(\varphi_{\mathrm{PS}}, \psi_{\mathrm{PS}})$ then $\mathcal{G}$ is a PCP-structure and if $\mathcal{G}' = (\mathbb{V}', \mathbb{D}')$ is an unlabelled PCP-structure then there is labelling function $L'$ such that $(\mathbb{V}', \mathbb{D}', L')$ satisfies $(\varphi_{\mathrm{PS}}, \psi_{\mathrm{PS}})$.

*Proof.* Assume that $\mathcal{G}$ satisfies $(\varphi_{\mathrm{PS}}, \psi_{\mathrm{PS}})$. As $\varphi_{\mathrm{CL}}$ occurs as a conjunct in $\varphi_{\mathrm{PS}}$ and $\psi_{\mathrm{CL}}$ in $\psi_{\mathrm{PS}}$ Lemma 3 implies that $\mathcal{G}$ is a chain-ladder. Let $\mathbb{V}_1 \cup \mathbb{V}_2$ be the ladder and $\mathbb{V}_3$ the chain.

The subformula $\mathrm{exactly\_one}(\mathbb{F})$ and $\mathrm{colour}(\mathbb{F})$ in combination with $(\bigwedge_{\mathbb{C}_3} \neg c_i \rightarrow \bigwedge_{\mathbb{F}} \neg(c_i, d))$ imply that a node is of colour $c_i$ if and only if it is of exactly one color $(c_i, d)$. From the arguments of Lemma 3 we know that each node $v$ has exactly one colour $c_i$ and, thus, $v$ also has a corresponding direction $d \in \{L, M, R\}$. The subformulas $((c_i, s) \rightarrow (c_i, L) \wedge \odot(c_i, M) = 1)$ and $((c_i, m) \rightarrow \bigvee_{\mathbb{F}} (c_i, d) \wedge \bigwedge_{d' \neq d} \odot(c_i, d') = 1)$ imply that start node of chain $\mathbb{V}_i$ has direction $L$ and its neighbour $M$ and that the neighbours of each middle node of direction $d$, characterised by colour $(c_i, m)$, must have directions $d-1$ and $d+1$. In combination, this implies that each chain $\mathbb{V}_1, \mathbb{V}_2$ and $\mathbb{V}_3$ is coloured from start to end with the pattern $(L, M, R)^*$.

The subformulas $(\bigwedge_{\{c_1, c_2\}} c_i \rightarrow \odot c_3 \le 1)$ and $(c_3 \rightarrow \odot c_1 = 1 \wedge \odot c_2 = 1)$ imply that nodes from ladder $\mathbb{V}_1 \cup \mathbb{V}_2$ have at most one neighbour from $\mathbb{V}_3$ and each node from chain $\mathbb{V}_3$ has exactly one neighbour from $\mathbb{V}_1$ and one from $\mathbb{V}_2$. Consider the dimensions $(d, c_i, \mathrm{id})$. First, the subformula $(\neg(c_3, d) \rightarrow \bigwedge_{\{c_1, c_2\}} (d, c_i, \mathrm{id}) = 0)$ and the conditions of $\varphi_{\mathrm{CL}}$ imply that dimension $(d, c_i, \mathrm{id})$ of node $v$ are nonzero only if $v$ is from $\mathbb{V}_3$ and of direction $d$. The subformula $((c_3, d) \rightarrow \bigwedge_{\{c_1, c_2\}} \odot(c_i, \mathrm{id}) = (d, c_i, \mathrm{id}))$ leads to the case that each node $v \in \mathbb{V}_3$ of direction $d$ has stored the id of its one neighbour from $\mathbb{V}_1$ in $(c_1, d, \mathrm{id})$ and the id of its one neighbour from $\mathbb{V}_2$ in $(c_2, d, \mathrm{id})$. Now, the subformulas $((c_3, d) \rightarrow \bigwedge_{\{c_1, c_2\}} \odot(d-1, c_i, \mathrm{id}) \le (d, c_i, \mathrm{id}))$ and $((d, c_i, \mathrm{id}) \le \odot(d+1, c_i, \mathrm{id}))$ imply the main property of a PCP-structure, namely that the connections between $\mathbb{V}_3$ and $\mathbb{V}_1$ as well as $\mathbb{V}_2$ are not intersecting. Note that $\le$ is sufficient as each node from $\mathbb{V}_1$ and $\mathbb{V}_2$ can have at most 1 neighbour from $\mathbb{V}_3$. $\quad\square$

Finally, we are set to prove that DGLP is undecidable. Let $P = \{(\alpha_1, \beta_1), \ldots, (\alpha_k, \beta_k)\}$ be a PCP instance over alphabet $\Sigma = \{a, b\}$ and let $\tilde{m} = \max(\bigcup_{i=1}^{k} \{|\alpha_i|, |\beta_i|\})$. Let $\mathbb{B} = \{(c_i, d, a, j), (c_i, d, b, j) \mid c_i \in \{c_1, c_2\}, d \in \{L, M, R\}, j \in \{0, \ldots, \tilde{m} - 1\}\}$ and $\mathbb{S} =$

$\{(c_i, d, p, j) \mid c_i \in \{c_1, c_2\}, d \in \{L, M, R\}, p \in \{1, \ldots, k, e, \bot\}, j \in \{0, \ldots, \tilde{m}\}\}$ colours. Additionally, we define $\mathbb{S}^\top = \mathbb{S} \setminus \{(c_i, d, \bot, j) \mid (c_i, d, \bot, j) \in \mathbb{S}\}$, $\mathbb{S}^0 = \{(c_i, d, p, 0) \mid (c_i, d, p, 0) \in \mathbb{S}\}$ and $\mathbb{B}^0 = \{(c_i, d, p, 0) \mid (c_i, d, p, 0) \in \mathbb{B}\}$. We define the following DGLP $(\varphi_P, \psi_P)$ over the variables $\mathrm{Var}_{\mathsf{PS}} \cup \{\mathrm{x}_c \mid c \in \mathbb{B} \cup \mathbb{S}\}$:

$$\varphi_P := \varphi_{\mathrm{cond}} \wedge (\bigwedge_{\{c_1, c_2\}} c_i \rightarrow \bigwedge_{\mathbb{M} \in \{\mathbb{B}^0, \mathbb{S}^0\}} \mathrm{exactly\_one}(\mathbb{M})) \wedge \varphi_{\mathsf{PS}} \wedge \mathrm{colour}(\mathbb{B} \cup \mathbb{S})$$

$$\begin{aligned}
\varphi_{\mathrm{cond}} := & \wedge \bigwedge_{\mathbb{F}} (\neg(c_i, d) \rightarrow \bigwedge_{\mathbb{B} \cup \mathbb{S}} \neg(c_i, d, p, j)) \\
& \wedge \bigwedge_{\mathbb{F}} (c_i, d) \rightarrow \bigwedge_{\mathbb{B}, j < \tilde{m}-1} (c_i, d, p, j+1) = \odot(c_i, d+1, p, j)) \\
& \wedge \bigwedge_{\mathbb{F}} (c_i, d) \rightarrow \bigwedge_{\mathbb{S}, j < \tilde{m}} (c_i, d, p, j+1) = \odot(c_i, d+1, p, j)) \\
& \wedge \bigwedge_{\{c_1, c_2\}} (c_i, e) \rightarrow \bigwedge_{\mathbb{B} \cup \mathbb{S}, p \neq e} \neg(c_i, d, p, j) \wedge \bigvee_{\mathbb{F}} (c_i, d, e, 0) \\
& \wedge \bigwedge_{\{c_1, c_2\}} \neg(c_i, e) \rightarrow \bigwedge_{\mathbb{F}} \neg(c_i, d, e, 0) \\
& \wedge \bigwedge_{\mathbb{S}^\top, p \neq e} (c_1, d, p, 0) \rightarrow \odot c_3 = 1 \wedge (\bigwedge_{j=0}^{|\alpha_p|-1} (c_1, d, \alpha_p[j], j) \wedge (c_1, d, \bot, j)) \\
& \qquad\qquad \wedge \bigvee_{p'=1}^{k} (c_1, d, p', |\alpha_p|) \vee (c_1, d, e, |\alpha_p|) \\
& \wedge \bigwedge_{\mathbb{S}^\top, p \neq e} (c_2, d, p, 0) \rightarrow \odot c_3 = 1 \wedge (\bigwedge_{j=0}^{|\beta_p|-1} (c_2, d, \beta_p[j], j) \wedge (c_2, d, \bot, j)) \\
& \qquad\qquad \wedge \bigvee_{p'=1}^{k} (c_2, d, p', |\beta_p|) \vee (c_2, d, e, |\beta_p|) \\
& \wedge \bigwedge_{\{c_1, c_2\}} (c_i, s) \rightarrow \bigvee_{\mathbb{S}^\top} (c_i, d, p, 0) \\
& \wedge \bigwedge_{\{L, M, R\}} ((c_1, d) \rightarrow (c_1, d, a, 0) = \odot(c_2, d, a, 0) \wedge (c_1, d, b, 0) = \odot(c_2, d, b, 0)) \\
& \wedge c_3 \rightarrow (\bigwedge_{\mathbb{S}^0} \bigwedge_{d' \in \{L, M, R\}} \odot(c_1, d, p, 0) = \odot(c_2, d', p, 0)) \wedge \bigvee_{\mathbb{S}^\top} \odot(c_1, d, p, 0) = 1 \\
\psi_P := & \psi_{\mathsf{PS}}
\end{aligned}$$

**Proof of Theorem 3.** We prove this via reduction from PCP. Let $P = \{(\alpha_1, \beta_1), \ldots, (\alpha_k, \beta_k)\}$ and $(\varphi_P, \psi_P)$ be like above. Assume that $(\varphi_P, \psi_P)$ is satisfied by $\mathcal{G}$.

From the describtion above, we can see that $\varphi_{\mathsf{PS}}$ and $\psi_{\mathsf{PS}}$ are conjunctive subformulas of $\varphi_P$ respectively $\psi_P$. Therefore, Lemma 4 implies that $\mathcal{G}$ is a PCP-structure. Let $\mathbb{V}_1 \cup \mathbb{V}_2$ be the ladder and $\mathbb{V}_3$ the additional chain in $\mathcal{G}$. In addition to the colours resulting from $\varphi_{\mathsf{PS}}$, the subformulas $\mathrm{colour}(\mathbb{B} \cup \mathbb{S})$ and $(\bigwedge_{\{c_1, c_2\}} c_i \rightarrow \bigwedge_{\mathbb{M} \in \{\mathbb{B}^0, \mathbb{S}^0\}} \mathrm{exactly\_one}(\mathbb{M}))$ ensure that $\mathbb{B}$ and $\mathbb{S}$ are colours and that each ladder node has exactly one colour from $\mathbb{B}^0 \subset \mathbb{B}$ and $\mathbb{S}^0 \subset \mathbb{S}$. The idea of these colours is the following: a colour $(c_i, d, p, j) \in \mathbb{B}$ with $p \in \{a, b\}$ and $j \in \{0, \ldots, \tilde{m}-1\}$ represents the symbol ($a$ or $b$) of a node in distance $j$ of a node coloured with $(c_i, d)$. Similarly, colour $(c_i, d, p, j)$ with $p \in \{1, \ldots, k, \bot, e\}$ and $j \in \{0, \ldots, \tilde{m}\}$ represents that a node in distance $j$ of a node coloured with $(c_i, d)$ is the start of tilepart $\alpha_p$ if $i = 1, p \neq \bot, e$ and $\beta_p$ if $i = 2, p \neq \bot, e$. In case of $p = e$ the node in distance $j$ is the end node of chain $\mathbb{V}_i$ and $p = \bot$ is a placeholder for nodes which are neither a start of some tilepart nor the end node. The case $j = 0$ is interpreted as its own symbol or start of a tile part.

We argue how $\varphi_P$ ensures the above mentioned properties of colours $(c_i, d, p, j) \in \mathbb{B} \cup \mathbb{S}$. The subformula $(\neg(c_i, d) \rightarrow \bigwedge_{\mathbb{B} \cup \mathbb{S}} \neg(c_i, d, p, j))$ ensures that a node of some colour $(c_i, d, p, j)$ must also be of colour $(c_i, d)$. Especially, this implies that nodes from chain $\mathbb{V}_3$ do not have any colour $(c_i, d, p, j)$. The subformulas $((c_i, d) \rightarrow \bigwedge_{\mathbb{B}, j < \tilde{m}-1} (c_i, d, p, j+1) = \odot(c_i, d+1, p, j))$ and $((c_i, d) \rightarrow \bigwedge_{\mathbb{S}, j < \tilde{m}} (c_i, d, p, j+1) = \odot(c_i, d+1, p, j))$ ensure that a node with colour $(c_i, d)$ stores the information $p, j$ of its $(c_i, d+1)$ neighbour in form of its own colour $(c_i, d, p, j+1)$. Note that, each chain is labeled wird $L, M, R, L, \ldots$ from start to end and, thus, the $d+1$ neighbour is the *right* neighbour in the sense that its nearer to the end node $v_e$. To understand how this leads to the case that each node on chain $\mathbb{V}_i$ stores the information of its $\tilde{m}$ right neighbours, we argue beginning from end $v_e$ of chain $\mathbb{V}_i$. Subformula $((c_i, e) \rightarrow \bigwedge_{\mathbb{B} \cup \mathbb{S}, p \neq e} \neg(c_i, d, p, j) \wedge \bigvee_{\mathbb{F}} (c_i, d, e, 0))$ ensures that $v_e$ only has colour $(c_i, d, e, 0)$. That $d$ matches its colour $(c_i, d)$ is ensured by $\neg(c_i, d) \rightarrow \cdots$. Therefore, its only and left neighbour $v$ must have colour $(c_i, d-1, e, 1)$ plus its own additional colours with $j = 0$. Now, the left neighbour $v'$ of $v$ must have colours $(c_1, d-2, e, 2)$, the colours equivalent to $v$ with $j = 1$ and its own colours with $j = 0$ and so on. As the maximum $j$ in case of a colour from $\mathbb{S}$ is $\tilde{m}$, tilepart start, end or $\bot$ colours are stored in nodes up to distance $\tilde{m}$ to the left of the original node. The same holds for colours from $\mathbb{B}$ with distance $\tilde{m} - 1$.

We are set to argue that $\mathcal{G}$ encodes a solution $I$ of $P$. The subformula $((\bigwedge_{\mathbb{S}^\top, p \neq e}(c_1, d, p, 0) \to \cdots \wedge (\bigwedge_{j=0}^{|\alpha_p|-1}(c_1, d, \alpha_p[j], j) \wedge (c_1, d, \bot, j)) \wedge \bigvee_{p'=1}^k (c_1, d, p', |\alpha_p|)) \vee (c_1, d, e, |\alpha_p|))$ ensures that for each node from $\mathbb{V}_1$ that is a tilepart start for some $\alpha_p$ that $\alpha_p$ is written to the right without a next tilepart starting $((\bigwedge_{j=0}^{|\alpha_p|-1}(c_1, d, \alpha_p[j], j) \wedge \wedge (c_1, d, \bot, j)))$ and that after $\alpha_p$ is finished that either the next tile part starts or the chain ends $(\bigvee_{p'=1}^k (c_1, d, p', |\alpha_p|) \vee (c_1, d, e, |\alpha_p|))$. Analogous conditions are ensured for nodes from $\mathbb{V}_2$ by the subformula $((\bigwedge_{\mathbb{S}^\top, p \neq e}(c_2, d, p, 0) \to \cdots \wedge (\bigwedge_{j=0}^{|\beta_p|-1}(c_2, d, \beta_p[j], j) \wedge (c_2, d, \bot, j)) \wedge \bigvee_{p'=1}^k (c_2, d, p', |\beta_p|) \vee (c_2, d, e, |\beta_p|))$. Now, the subformula $(\bigwedge_{\{c_1, c_2\}}(c_i, s) \to \bigvee_{\mathbb{S}^\top}(c_i, d, p, 0))$ ensures that the start nodes of $\mathbb{V}_1$ and $\mathbb{V}_2$ correspond to a tilepart start. In combination with the previous conditions, this ensures that chains $\mathbb{V}_1$ and $\mathbb{V}_2$ are coloured with words $w_\alpha$ and $w_\beta$ corresponding to sequences $\alpha_{i_1} \cdots \alpha_{i_h}$ and $\beta_{j_1} \cdots \beta_{j_l}$.

It is left to argue that $w_\alpha = w_\beta$ and that $i_1 \cdots i_h = j_1 \cdots j_l$. The first equality is ensured by $((c_1, d) \to (c_1, d, a, 0) = \odot(c_2, d, a, 0) \wedge (c_1, d, b, 0) = \odot(c_2, d, b, 0))$ and the fact that $\mathbb{V}_1 \cup \mathbb{V}_2$ is a ladder. The second equality is ensured by $(c_3 \to (\bigwedge_{\mathbb{S}^0} \bigwedge_{d' \in \{L, M, R\}} \odot(c_1, d, p, 0) = \odot(c_2, d', p, 0)) \wedge \bigvee_{\mathbb{S}^\top} \odot(c_1, d, p, 0) = 1)$, $((c_1, d, p, 0) \to \odot c_3 = 1 \wedge \cdots)$ and $((c_2, d, p, 0) \to \odot c_3 = 1 \wedge \cdots)$ and the fact that $\mathcal{G}$ is a PCP-structure which means that connections between $\mathbb{V}_3$ and $\mathbb{V}_1$ respectively $\mathbb{V}_2$ are not intersecting. Therefore, the sequence $i_j \cdots i_h = j_1 \cdots j_l$ is a solution for $P$ which implies that $P$ is solvable.

The vice-versa direction, namely that if $P$ is solvable then $(\varphi_P, \psi_P)$ is satisfiable, is argued easily: If $P$ is solvable then there is a solution $I$. Figure 3 indicates how to encode $I$ as a PCP-structure $\mathcal{G}$. Note that in contrast to the visualisation, the encoding characterized by $\varphi_{\text{PS}}$ demands that the end nodes of chain $\mathbb{V}_1$ and $\mathbb{V}_2$ are not part of solution $I$. Lemma 4 states that for each unlabeled PCP-structure there is a labeling function $L'$ such that $\mathcal{G}$ satisfies $(\varphi_{\text{PS}}, \psi_{\text{PS}})$. Therefore, if we take a matching PCP-structure $\mathcal{G}$ without labels, label it with $L'$ and then extend $L'$ with the colours $(c_i, d, p, j)$ according to $I$ and the arguments above we get that $\mathcal{G}$ satisfies $(\varphi_P, \psi_P)$. $\square$

We can see from the definitions of $\varphi_{\text{CL}}$, $\varphi_{\text{PS}}$ and $\varphi_P$ and corresponding graph conditions that they belong to the DGLP fragment of GLP. This proves the statement of Corollary 3.

## B.2 Proving that DGLP is reducible to GCP

In the proof of Theorem 4 we claimed the following properties of $\langle x \leq m \rangle$ and $\langle x \in \mathbb{M} \rangle$ gadgets.

**Lemma 5.** Let $r \in \mathbb{R}$ and $(r_1, \ldots, r_k) \in \mathbb{R}^k$ for some $k$. It holds that $\langle r \leq m \rangle = 0$ if and only if $r \leq m$ and $\langle r \in \mathbb{M} \rangle = 0$ if and only if $r \in \mathbb{M}$. Furthermore, gadgets $\langle x \leq m \rangle$ and $\langle x \in \mathbb{M} \rangle$ are positive and $\langle x \leq m \rangle$ is upwards bounded.

*Proof.* The properties of $\langle x \leq m \rangle$ are straightforward implications of its functional form.

Next, we prove that $\langle r \in [m; n] \rangle = 0$ if and only if $r \in [m; n]$. Assume that $r \in [m; n]$. It follows that the output of each inner ReLU node is 0 and therefore $\langle r \in [m; n] \rangle = 0$. Next, assume $r < m$. It follows that $\text{re}(m - x) > 0$ and as the value of all other inner ReLU nodes must be greater or equal to 0 it follows that $\langle r \in [m; n] \rangle > 0$. The case $r > n$ is argued analogously as $\text{re}(x - n) > 0$.

Consider the $\langle x \in \mathbb{M} \rangle$ gadget and assume that $r \in \mathbb{M}$. It clearly holds that $r \in [i_1; i_k]$ and therefore that $\text{re}(\langle x \in [i_1; i_k] \rangle) = 0$. Furthermore, w.l.o.g. let $r = i_l$ for some $1 \leq l < k$. Then, it follows that $\frac{(i_{l+1}-i_l)}{2} = \text{re}(r - \frac{i_l+i_{l+1}}{2}) + \text{re}(\frac{i_l+i_{l+1}}{2} - r)$ and $\frac{(i_{j+1}-i_j)}{2} < \text{re}(r - \frac{i_j+i_{j+1}}{2}) + \text{re}(\frac{i_j+i_{j+1}}{2} - r)$ for $j \neq l$ and, thus, the inner sum is equal to 0 as well. Now, assume that $r \notin \mathbb{M}$. If $r < i_1$ or $r > i_k$ it follows that $\text{re}(\langle x \in [i_1; i_k] \rangle) > 0$. If $r \in [i_1; i_k]$ it must be the case that $r \in (i_j; i_{j+1})$ for some $i \leq j < k$ and therefore that $\text{re}(\frac{(i_{j+1}-i_j)}{2} - (\text{re}(x - \frac{i_j+i_{j+1}}{2}) + \text{re}(\frac{i_j+i_{j+1}}{2} - x))) > 0$. That the gadget $\langle x \in \mathbb{M} \rangle$ is positive is obvious as the outermost function is ReLU. $\square$

## B.3 Proving the Tree-Model Property of Node-Classifier MPNN Over Bounded Graphs

In the proof of Theorem 2 we claim that node-classifier MPNN have the tree-model property. We formally prove this statement in the following.

Let $\mathcal{G} = (\mathbb{V}, \mathbb{D}, L)$ be a graph and $v \in \mathbb{V}$. The set of *straight i-paths* $\mathbb{P}_v^i$ of $v$ is defined as $\mathbb{P}_v^0 = \{v\}$, $\mathbb{P}_v^1 = \{vv' \mid (v, v') \in \mathbb{D}\}$ and $\mathbb{P}_v^{i+1} = \{v_0 \cdots v_{i-1} v_i v_{i+1} \mid v_0 \cdots v_{i-1} v_i \in \mathbb{P}_v^i, (v_i, v_{i+1}) \in \mathbb{D}, v_{i-1} \neq v_{i+1}\}$. For some path $p = v_0 \cdots v_l$ we define $p_i$ for $i \leq l$ as the prefix $v_0 \cdots v_i$. Furthermore, let $\tilde{\mathbb{P}}_v^i = \{(v_0, p_0)(v_1, p_1) \cdots (v_i, p_i) \mid p = v_0 \cdots v_i \in \mathbb{P}_v^i\}$ and let $\tilde{\mathbb{P}}_v^i(j) = \{(v_j, p_j) \mid (v_0, p_0) \cdots (v_j, p_j) \cdots (v_i, p_i) \in \tilde{\mathbb{P}}_v^i\}$. Let $N$ be a node-classifier MPNN. We denote the value of $v$ after the application of layer $l_i$ with $N^i(\mathcal{G}, v)$ for $i \geq 1$. Furthermore, let $N^0(\mathcal{G}, v) = L(v)$ for each node $v$.

**Lemma 6.** Let $(N, \varphi, \psi) \in \mathrm{ORP}_{\mathsf{node}}(\Phi_{\mathsf{bound}}, \Psi_{\mathsf{conv}})$ where $N$ has $k'$ layers and $\varphi \in \Phi_{\mathsf{bound}}^{d,k''}$. The ORP $(N, \varphi, \psi)$ holds if and only if there is a $d$-tree $\mathcal{B}$ of depth $k = \max(k', k'')$ with root $v_0$ such that $(\mathcal{B}, v_0) \models \varphi$ and $N(\mathcal{B}, v_0) \models \psi$.

*Proof.* Assume that $(N, \varphi, \psi)$ is as stated above. The direction from right to left is straightforward. Therefore, assume that $(N, \varphi, \psi)$ holds. By defintion, there exists a $d$-graph $\mathcal{G} = (\mathbb{V}, \mathbb{D}, L)$ with node $w$ such that $(\mathcal{G}, w) \models \varphi$ and $N(\mathcal{G}, w) \models \psi$. Let $\mathcal{B} = \{\mathbb{V}_{\mathcal{B}}, \mathbb{D}_{\mathcal{B}}, L_{\mathcal{B}}, (w, w)\}$ be the tree with node set $\mathbb{V}_{\mathcal{B}} = \bigcup_{i=0}^{k} \tilde{\mathbb{P}}_w^k(i)$. The set of edges $\mathbb{D}_{\mathcal{B}}$ is given in the obvious way by $\{((v, p), (v', pv')) \mid (v, p), (v', pv') \in \mathbb{V}_{\mathcal{B}}\}$ and closed under symmetrie. Note that from the definition of $\tilde{\mathbb{P}}_w^k$ follows that $\mathcal{B}$ is a well-defined $d$-tree of depth $k$. The labeling function $L_{\mathcal{B}}$ is defined such that $L_{\mathcal{B}}((v, p)) = L(v)$ for all $(v, p) \in \mathbb{V}_{\mathcal{B}}$. From its construction follows that $(\mathcal{B}, (w, w)) \models \varphi$.

We show that it holds that $N^{k'}(\mathcal{B}, (w, w)) = N^{k'}(\mathcal{G}, w)$ which directly implies that $N(\mathcal{B}, (w, w)) = N(\mathcal{G}, w)$. We do this by showing the following stronger statement for all $j = 0, \ldots, k'$ via induction: for all $(v, p) \in \tilde{\mathbb{P}}_w^{k'}(k' - i)$ with $k' \geq i \geq j$ holds that $N^j(\mathcal{B}, (v, p)) = N^j(\mathcal{G}, v)$. The case $j = 0$ is obvious as $L_{\mathcal{B}}$ is defined equivalent to $L$. Therefore assume that the statement holds for $j \leq k' - 1$ and all $(v, p) \in \tilde{\mathbb{P}}_w^{k'}(k' - i)$ with $k' \geq i \geq j$. Consider the case $j + 1$ and let $(v, p) \in \tilde{\mathbb{P}}_w^{k'}(k' - i)$ for some $k' \geq i \geq j$. We argue that for each $(v', p') \in \mathrm{Neigh}((v, p))$ it follows that $v' \in \mathrm{Neigh}(v)$ such that $N^j(\mathcal{B}, (v', p')) = N^j(\mathcal{G}, v')$ and vice-versa. Let $(v', p') \in \mathrm{Neigh}((v, p))$. By definition of $\mathbb{D}_{\mathcal{B}}$, $i \geq 1$ and the fact that all $p \in \mathbb{P}_w^{k'}$ are straight follows that $p' = pv'$ or $p = p'v$ but not both. Therefore, either $(v', p') \in \tilde{\mathbb{P}}_w^{k'}(k' - (i - 1))$ or $(v', p') \in \tilde{\mathbb{P}}_w^{k'}(k' - (i + 1))$ and, thus, $(v', v) \in \mathbb{D}$ or $(v, v') \in \mathbb{D}$ which in both cases means $v' = \mathrm{Neigh}(v)$. The induction hypothesis implies that $N^j(\mathcal{B}, (v', p')) = N^j(\mathcal{G}, v')$. As these arguments hold for all $(v', p') \in \mathrm{Neigh}((v, p))$ we get that $\sum_{\mathrm{Neigh}((v,p))} N^j(\mathcal{B}, (v', p')) \leq \sum_{\mathrm{Neigh}(v)} N^j(\mathcal{G}, v')$. The vice-versa direction is argued analogously which then implies that $\sum_{\mathrm{Neigh}((v,p))} N^j(\mathcal{B}, (v', p')) = \sum_{\mathrm{Neigh}(v)} N^j(\mathcal{G}, v')$. From the induction hypothesis we get that $N^j(\mathcal{B}, (v, p)) = N^j(\mathcal{G}, v)$. Then, the definition of a MPNN layer implies that $N^{j+1}(\mathcal{B}, (v, p)) = N^{j+1}(\mathcal{G}, v)$. Therefore, the overall statement holds for all $j \leq k'$ and by taking $j = i = k'$ we get the desired result. $\square$

