# OpenReview forum: "Fundamental Limits in Formal Verification of Message-Passing Neural Networks"
_ICLR.cc/2023/Conference — ICLR 2023 poster_

### Official Review · Reviewer_UR4T · 2022-10-14

**Confidence:** 4
**Correctness:** 2
**Technical Novelty And Significance:** 2
**Empirical Novelty And Significance:** Not applicable
**Recommendation:** 3

**Clarity, Quality, Novelty And Reproducibility:**

The presentation is good and the work is original for the particular problem it is studying, to my best knowledge. But the problem seems to be minor and the conclusions are not practically useful for now. Reproducibility is not applicable for this purely theoretical paper.


**Strength And Weaknesses:**

Strengths:
- This paper has theoretical results showing the impossibility of generally verifying MPNN, and a theoretical construction showing the possibility of verifying MPNN when the degree of the graph is bounded.

Weaknesses:
- The problem considered in this paper has little importance and the results are not useful in practice. For verification in practice, we not only want decidability, but also computationally feasibility. NP-hardness of verifying NN is known. Therefore, considering the decidability of verification for graphs with unbounded size and degrees is not important, as computational complexity would play a more important role for larger graphs, which limits complete verification. Decidability is minor here, which is not the major factor hindering formal verification.
- In practice, it is reasonable and acceptable for verification to only support specifications with some constraints, such as the size and degree of the graphs. While Theorem 1 says the impossibility of general verification, it would not fundamentally limit verification in practice.
- The contribution of Theorem 2 is not significant. It only proves the possibility of verification when the degree is bounded, without proposing a computationally feasible algorithm (the construction seems to have an exponential complexity and thus it does not automatically imply a computationally feasible algorithm). On regular NNs, there already exist many concrete algorithms for formal verification. Then the contribution here is relatively insignificant.


**Summary Of The Paper:**

This paper studies the formal output reachability verification of message passing neural networks (MPNN). The paper claims that the verification is impossible if for general graphs (unbounded size, nontrivial degree and node labels). But the verification is possible if there is a bound on the degree of the graph.



**Summary Of The Review:**

This work has theoretical merits on showing the impossibility of general GNN verification. But the problem is minor. The “fundamental limit” is not a critical limit in practice and is not the critical factor hindering the development of formal verifiers, for two reasons: 1) computational complexity plays a more important role given the NP-hardness of NN verification; 2) constraints can be added on graphs in practice, which makes the verification possible. The second point corresponds to Theorem 2 in the paper, but the paper does not provide a concrete and computationally feasible algorithm while only proving the possibility of verification, and thus the contribution is not significant.

Updates:

I am recommending a rejection because I think the contributions in this paper are insufficient, not because this is purely a theory paper. As I have said, NP-hardness in verification is known and is already fundamentally limiting the verification [1, 2]. Especially on verification for graphs with unbounded degree/size, NP-hardness is a stronger limit that can hinder exact verification in practice for unbounded cases. That’s also partly why I said verification for the unbounded case is not important as we have already known NP-hardness on verification so I would not pursue generalizing verification to the unbounded case in practice. Therefore, for now, the limit revealed by Theorem 1 (decidability) is not an important limit compared to NP-hardness done by previous works and the contribution is minor. However, it could potentially become more significant if the paper can prove that the verification is not NP-hard in the special case for MPNN considered in this paper, to eliminate the limit revealed in previous works so that decidability could really be interesting and important. Theorem 2 only showing the possibility without a computationally feasible method is not making a sufficient contribution either.

Actually the failure to support that decidability is "worse than NP-hard" ([unconvincingly claimed by the authors](https://openreview.net/forum?id=WlbG820mRH-&noteId=LXeVO_iFIJ)) poses a significant weakness of this paper. This paper provides no evidence to support such a claim. Without it, given the existing works on NP-hardness [1 2], showing the undecidability is insufficient to support the significance of this work. See [my previous thread](https://openreview.net/forum?id=WlbG820mRH-&noteId=u3TrgCTLzp) on why I disagree decidability is worse than NP-hard.

The authors refuse to compare with existing works on regular NN, by saying this paper works on GNN but not regular NN. GNN is an extension of regular NN. There are works about NP-hardness of verification on regular NN and this paper needs to show that why their results don't apply to this paper in a formal way, e.g., by proving that the verification in their case is not NP-hard, which is missing in this paper and I am not sure if it is actually possible to prove that. The authors are responsible for ensuring their claims are sufficiently supported.

Overall, there is a significant lack of evidence supporting the significance of the work. I believe this paper is not ready for publication at ICLR.

[1] Katz, G., Barrett, C., Dill, D. L., Julian, K., & Kochenderfer, M. J. (2021). Reluplex: a calculus for reasoning about deep neural networks. Formal Methods in System Design, 1-30.

[2] Sälzer, Marco, and Martin Lange. "Reachability is NP-Complete Even for the Simplest Neural Networks." International Conference on Reachability Problems. Springer, Cham, 2021.

My latest discussion [post1](https://openreview.net/forum?id=WlbG820mRH-&noteId=u3TrgCTLzp) and [post2](https://openreview.net/forum?id=WlbG820mRH-&noteId=sbBDJ-bL3L).

---

> ### Author Response · Authors · 2022-11-16
> **response regarding misunderstandings in your review and the practical relevance of our contribution (1)**
>
> First of all, we appreciate the time that went into writing this review. We feel your review needs a thorough response, which will be a bit extensive. For this, we apologize in advance.
>
> **Our response is twofold**:
> - To be able to adequately address the concerns raised regarding our submission,
> we must correct some misleading/incorrect statements made in your review. This will be the first part of our response.
> - After clearing up several misunderstandings, we discuss concerns, which seem valid at first glance,
> regarding the practical impact of our impossibility result(s). This will be the second part of our response.
>
> ### First Part: Clearing Up Several Misunderstandings (A)
> > *"Do people use graphs with unbounded size in practice?"; "How can a system have an unbounded size?"; "At least, there should be some limit on the storage quota .."; "This paper considers an artificial scenario where the graph can have unbounded sizes, which is unrealistic."*
>
> These statements in your review show that there is a confusion between the terms "boundedness" and "finiteness" (or "unboundedness" and "infinity"). To clear this situation: at no point in our work do we assume that a graph is infinite which corresponds to the normal usage scenario of MPNN. (see the bottom of page 2 "... where V is a finite set of nodes, ..."). It seems that your claims about an unrealistic scenario are simply due to this misunderstanding and, thus, are misplaced. Please also see the second part of our [answer to reviewer 8dRF](https://openreview.net/forum?id=WlbG820mRH-&noteId=F4kat40Vglw). To avoid this misunderstanding in the future, we have revised corresponding wordings in the entire work. See the revised version of our paper.
>
> > *"This paper theoretically showed that the unbounded MPNN cannot be formally verified while verifying MPNN with bounded sizes is possible."*
>
> We are confused by this statement. In no part of our work do we define or use the term "bounded MPNN" or "unbounded MPNN". Therefore, we assume that you are talking about "(un)bounded graph classes" when referring to this statement.
>
> >*"The paper claims that MPNN is generally not verifiable if the size of the graph can be unbounded, but the verification becomes possible when the size of the graph is bounded.";
> "This paper theoretically showed that the unbounded MPNN cannot be formally verified while verifying MPNN with bounded sizes is possible."; "Theorem 2 says ORP is formally verifiable when the graph has a bounded size."; "...results would be totally different in more realistic scenarios where the size is bounded, as shown in Theorem 2."*
>
> Here seems to be another misunderstanding: at no point in this paper do we assume a bound on the size of graphs. The only property we assume to be bounded is the degree of graphs (Thm. 2). See the paragraph which shortly introduces  $\Phi_\text{bound}$ in Sect. 3 on page 4 and the corresponding paragraph of Appendix A on page 13.
>
> > *"Theorem 1 is only for an artificial scenario which is unrealistic and has been overturned by Theorem 2 when a realistic assumption is added (the graph should have a bounded size)"*
>
> One can in no way take the result on Thm. 2 to invalidate Thm. 1, since they deal with different settings: Thm. 1 is about graph-classifier MPNN and Thm. 2 is about node-classifier MPNN.  An informal description of the difference between the two models is given in the [answer to reviewer 5S64](https://openreview.net/forum?id=WlbG820mRH-&noteId=d2NCG8YfoX). The formal definition can be found in Sect. 2 page 2 in the paragraph about MPNN. Further, we refer to the definitions of $\Phi_\text{unb}$ and  $\Phi_\text{bound}$ (Sect. 3 and Appendix A) as well as the last paragraph of Sect. 6 on page 9.
>
> To informally summarize both results: Theorem 1 states that as soon as we consider finite graphs with sufficiently large degree (at least 4) and expressive labels (vectors with medium number of dimensions), formal verification of graph-classifier MPNN is no longer possible. Theorem 2 states that as soon as we place a bound on the degree of the considered finite graphs, formal verification of node-classifier MPNN is possible. Again, we would like to stress that we never talk about bounding the size of considered graphs.

---

> > ### Author Response · Authors · 2022-11-16
> > **response regarding misunderstandings in your review and the practical relevance of our contribution (2)**
> >
> > ### First Part: Clearing Up Misunderstandings (B)
> >
> >
> > >*"The information conveyed by Theorem 2 is trivial."; "Theorem 2, ..., does not convey any nontrivial message, given the existence of many NN verifiers."; "The contribution of this theorem (Thm. 2) is trivial when it does not give any concrete verification algorithm."*
> >
> > The term "trivial" is being used in your review and comment in a non-standard way. It seems like what you mean may be "meaningless" rather than "trivial".
> >
> > First, we would like to point out that this work does not deal with classical NN, but with GNN. These models are completely different in terms of definition and domain. Therefore, the existence of (complete and sound) verification algorithm for classical NN does not make Thm. 2 meaningless or contradicts Thm. 1.
> >
> > Second, note that we *do* give a concrete algorithm for the verification of output reachability properties of node-classifier MPNN, a direct consequence of the proof technique used in Thm. 2. However, this algorithm can only be seen as a proof of concept. But this is not the point of this result and our submitted work in general.
> >
> > The point of the paper is to study the principle limits as to where automatic verification of MPNN is algorithmically possible (in a sound, complete and terminating way). Thm. 1 shows that there are impossible settings. The main contribution of Thm. 2 is then the identification of particular parameters in the problem description (bound on the graph degree and node- vs. graph-classification) which are essential for the possibility or impossibility of formal verification. Hence, Thm. 1 and Thm. 2  narrow down the position of the borderline between possibility and impossibility, and they only do so together.
> >
> > > *"Results and conclusions are not novel and ..."*
> >
> > We are confused by this bold statement as we are not aware of any work in this direction. If you know of any work that anticipates the results of this paper, we would be very grateful if you could refer to it.

---

> > > ### Author Response · Authors · 2022-11-16
> > > **response regarding misunderstandings in your review and the practical relevance of our contribution (3)**
> > >
> > > ### Second Part: Concerns About Practical Relevance
> > > Leaving aside the misconceptions above, your review hints at the question of practical relevance of Thm.1.
> > > We suspect this is comes from the idea that, once you assume a bound on the size of graphs, formal verification is possible and in practical applications there always has to be some bound.
> > >
> > > The point here is not whether an *a posteriori* bound exists regarding all inputs that have actually
> > > been given to an implementation of an algorithm, but whether such an *a priori* bound precludes
> > > the application of an algorithm to certain inputs.
> > >
> > > In detail, this means: for any bound $n$, you can do formal verification of MPNN over graphs of
> > > size at most $n$. (This is in fact a "trivial" result, as there are only finitely many (unlabeled)
> > > graphs for each bound n.) So let's say some practitioner has found a great method that works
> > > well in practice for graphs of size up to 1M nodes. But then maybe next year there are
> > > applications that require 2M nodes, or computer power has increased, or memory has become
> > > cheaper etc. Now what? The practitioner designs a new algorithm, and the game continues. Now
> > > Thm. 1 essentially says that this game will continue forever: for every algorithm (regardless
> > > of how well it works on MPNN over graphs up to a size of some $n$) there are inputs that will
> > > not be captured by *this* algorithm. And experience shows that, as time goes on, such inputs
> > > that have formerly been seen as out of reach are then becoming relevant (and, thus, from a
> > > practitioners point of view the scene is shifting, but the theoretician points at his theorems
> > > and says: "we don't need new theorems; we have already covered this case as well!")
> > >
> > > So Thm. 1 is indeed relevant in practice; it does not make a statement about nor contradict
> > > the existence of *one particular* verification method which has proved to be useful on certain
> > > benchmarks up to certain sizes. Instead, it puts a limit on what is possible in the design of
> > > verification algorithms *in general*. It is not about looking back and saying "my algorithm
> > > has only run on these x instances, so therefore there is a bound." It is about looking forward
> > > and say "I am designing an algorithm which will be correct for graphs." Thm. 1 says that in
> > > this particular setting, this cannot be done. The best that can be done is to say "I am designing
> > > an algorithm which will be correct for graphs of size up to $x$". So to answer your question: yes,
> > > people do use graphs of unbounded size in practice, by designing algorithms first and then testing
> > > empirically up to which bound they can be applied (and not vice-versa).
> > >
> > > So the consequence of the difference between an a priori bound and an a posteriori bound only
> > > comes out when unscrambling the meaning of boundedness, and the simplicity of the term itself
> > > may lead to a wrong intuition about its meaning. But it's not our invention; it is just the
> > > computer-science version of the hen-and-egg-problem - what was there first: the algorithm or
> > > its inputs? It is usually the algorithm (in which case unboundedness - as used here - does
> > > cover practical scenarios quite accurately). Imagine it is not about graphs and verification but
> > > lists and sorting. Clearly, every implementation of a sorting algorithm will only ever run on
> > > lists of bounded size. However, all the prominent and best sorting algorithms do not presuppose
> > > a-priori bounds on the length of the lists to be sorted. So it would be entirely correct to
> > > argue that "people use unbounded lists/graphs/... in practice" although it would be better to
> > > express this as "people who devise algorithms for practical use usually do so without explicit
> > > a priori bounds on their inputs".
> > >
> > > ---
> > >
> > > *Sorry again, for the lengthy response, but we felt it necessary to clear several things up.*

---

> ### Comment · Reviewer_UR4T · 2022-11-17
> **Concerns in my initial review are not due to misunderstandings mentioned in the reply**
>
> Thanks to the authors for the response.
>
> Apologize for my inaccurate wordings in my original review as pointed out by the authors. But I think these are not critical misunderstandings that would affect my review. If my wordings are adjusted according to the author response and some of the words are replaced, the arguments and concerns in my original review still exist. I disagree that `It seems that your claims about an unrealistic scenario are simply due to this misunderstanding and, thus, are misplaced`.
>
> >These statements in your review show that there is a confusion between the terms "boundedness" and "finiteness" (or "unboundedness" and "infinity"). To clear this situation: at no point in our work do we assume that a graph is infinite which corresponds to the normal usage scenario of MPNN. (see the bottom of page 2 "... where V is a finite set of nodes, ...")
>
> I didn’t say you are considering the graph to be infinite. I was saying in practice the graph should be bounded, so Theorem 1 is not practical. And about whether the bound is on the size of the graph or the degree, it does not affect my concerns in the review – graph size and degrees can all be bounded in practice.
>
> >One can in no way take the result on Thm. 2 to invalidate Thm. 1, since they deal with different settings
>
> I should not say Theorem 2 could overturn Theorem 1. But I mean people care more about the setting considered in Theorem 2 which has an opposite result compared to Theorem 1, so Theorem 1 is not useful, while Theorem 2 is not significant either.
>
>
> *Responding to author response "(1)" for now.*

---

> > ### Author Response · Authors · 2022-11-17
> > **short organizational note**
> >
> > To streamline the discussion, we included responding to this comment in our [response to its second part](https://openreview.net/forum?id=WlbG820mRH-&noteId=VDN4408lpB).

---

> ### Comment · Reviewer_UR4T · 2022-11-17
> **More clarification (responding to author reply (2))**
>
>
> >The term "trivial" is being used in your review and comment in a non-standard way. It seems like what you mean may be "meaningless" rather than "trivial".
>
> Trivial can refer to something that is “of little worth or importance”. See the [dictionary](https://www.merriam-webster.com/dictionary/trivial).
>
> >First, we would like to point out that this work does not deal with classical NN, but with GNN. These models are completely different in terms of definition and domain. Therefore, the existence of (complete and sound) verification algorithm for classical NN does not make Thm. 2 meaningless or contradicts Thm. 1.
>
> The contribution of Theorem 2 is still not significant. For regular NNs, there exist many concrete verification algorithms. I would expect those existing algorithms can also be applied to GNN as well in some way, although the quality may be not good enough. Since verifying regular NNs is possible, I would assume verifying GNN is also possible. In a trivial way, when the graph is bounded, the GNN can be expanded to a set of regular NNs. Then the contribution is too limited if a theorem simply says verification is possible without proposing a concrete and computationally feasible algorithm.
>
> >Second, note that we do give a concrete algorithm for the verification of output reachability properties of node-classifier MPNN, a direct consequence of the proof technique used in Thm. 2. However, this algorithm can only be seen as a proof of concept. But this is not the point of this result and our submitted work in general.
>
> The algorithm in the proof seems to have an exponential complexity and is thus not a computationally feasible one.
>
> >We are confused by this bold statement as we are not aware of any work in this direction. If you know of any work that anticipates the results of this paper, we would be very grateful if you could refer to it.
>
> Given the existence of many concrete verification algorithms on regular NNs, I don’t think only saying verifying GNN is possible or impossible without a concrete and computationally feasible algorithm is novel. The nonexistence of a previous paper having similar results as those in this paper does not automatically imply novelty.

---

> > ### Author Response · Authors · 2022-11-17
> > **response to clarifications of your result**
> >
> > *Note: We think the discussion would be easier to follow if you added your responses directly to the corresponding posts, not to your original review. To streamline the discussion, this comment regards the [one above](https://openreview.net/forum?id=WlbG820mRH-&noteId=JWSvPBJ69f) and [its first part](https://openreview.net/forum?id=WlbG820mRH-&noteId=ZslzjOv2omP).*
> >
> > ### Preliminary Informations
> >
> > Thanks, for continuing the discussion.
> > We think the most important explanation to help you clearing misunderstandings of our results is the one given [in the third part of our previous response](https://openreview.net/forum?id=WlbG820mRH-&noteId=2_6hlg4wF4E), which you seem to have skipped.
> >
> > ---
> >
> > ### Detailed Responses
> > Although we disagree with most of your justifications regarding the ambiguities from your original review, we fear that the discussion will derail. Therefore, we will only address the substantively relevant parts of your responses in detail and make only minor remarks on the remaining parts at the end.
> >
> > > For regular NNs, there exist many concrete verification algorithms. I would expect those existing algorithms can also be applied to GNN as well in some way, although the quality may be not good enough.
> >
> > Please, refrain from trying to refute our results with vague ideas, constructions, etc., which *"you would expect"* to work. We cannot respond to such arguments objectively.
> >
> > Nevertheless, we try to give a clear response: No, they cannot as the two models are very different, which we explained before. To do so generally, you would need to translate a GNN into an NN. This is possible for a pair of a fixed graph and fixed MPNN (using classical NN in their message or readout parts). Note that this is the core idea of the algorithm implied by Thm. 2. However, as we need to do this translation for every graph of the class of candidates individually, we need some argument that this loop terminates. For node-classifier MPNN over graph classes with bounded degree there is such an argument - the statement of Thm. 2.
> >
> > > Since verifying regular NNs is possible, I would assume verifying GNN is also possible.
> >
> > This is again a very vague statement. We think it is clear by now that classical NN and GNN are different models. In particular, they work over different inputs, the former works over vectors and the latter over labeled graphs. First, note that Thm. 1 is a clear counterexample for this statement.
> >
> > However, since you debate the result of Thm. 1, we also give you a more general explanation why such reasoning is wrong:
> > You claim that because "everything" is decidable with respect to the first model, it should also be so for the second. Presumably with the reasoning that these are quite similar after all (note: we disagree with this). There are a countless number of problems where a change in a single parameter, inconspicuous at first glance, can make the difference between decidability and undecidability. Take the PCP problem used in the technical parts of our work: it is decidable for unary alphabets (size 1) but undecidable for binary (size 2). Margenstern [1] did a survey on this topic, trying to pinpoint the fine line between decidability and undecidability for many problems from different disciplines.
> > Again, we would like to stress that this is also the case for NN and GNN, established by our work, although the difference between those models is in no sense minor.
> >
> > ---
> > ### Short Remarks
> >
> > > I didn’t say you are considering the graph to be infinite. I was saying in practice the graph should be bounded, ...
> >
> > If you are aware of the fact that all considered graphs are finite, we can not make sense of your statement of "the graph should be bounded". Each finite graph is trivially bounded, namely by its own size.
> >
> > > And about whether the bound is on the size of the graph or the degree, it does not affect my concerns in the review – graph size and degrees can all be bounded in practice.
> >
> > Exactly, those properties of graph classes *can* be bounded in practice, but they *do not have to be*. Now, what Thm. 1 states is that if they are not we can do no formal verification of graph-classifier MPNN.
> >
> > > But I mean people care more about the setting considered in Theorem 2 which has an opposite result compared to Theorem 1, ...
> >
> > Again, Thm. 1 and Thm. 2 are different settings, so they should not be compared. Furthermore, arguing that "people care more about the setting considered in ..." is subjective and calls for concrete evidence.
> >
> > > Trivial can refer to something that is “of little worth or importance”. See the [dictionary](https://www.merriam-webster.com/dictionary/trivial).
> >
> > We are conducting a technical/mathematical discussion. We took it as stated in item (b) of your reference.
> >
> > ---
> >
> > *[1] Maurice Margenstern, Frontier between decidability and undecidability: a survey, Theoretical Computer Science, Volume 231, Issue 2, 2000, Pages 217-251*

---

> > > ### Comment · Reviewer_UR4T · 2022-11-17
> > > **Acknowledging the theorectical contribution but the significance is limited**
> > >
> > >
> > > >...which you seem to have skipped.
> > >
> > > I didn’t skip it but would like to take more time to consider it after replying to your first two posts first (just like you replied to other reviewers at an earlier time, but it didn’t mean you skipped my review).
> > >
> > > Nevertheless, thanks for the explanation in the third part on why a more general verification may be desired but not possible according to Theorem 1. This may be an interesting theoretical contribution. But below, I would explain two reasons why I think Theorem 1 and 2 are not useful in practice:
> > >
> > > >This is again a very vague statement. We think it is clear by now that classical NN and GNN are different models. In particular, they work over different inputs, the former works over vectors and the latter over labeled graphs. First, note that Thm. 1 is a clear counterexample for this statement.
> > >
> > > Although they are different models and likely to need different verification techniques, I mean only proving that a verification is possible with a computationally infeasible construction does not make a sufficiently significant contribution, considering the existence of concrete verification algorithms on regular NNs.
> > >
> > > > You claim that because "everything" is decidable with respect to the first model, it should also be so for the second.
> > >
> > > This is a misunderstanding of my comments. I mean in practice, some constraints can naturally be added, including the size of the graph and degrees, and thus the decidability on graphs without these constraints is not important. Moreover, due to NP-hardness of verification for regular NN, such a decidability is less important, as computational cost would play a more important role. Due to practical constraints on the cost, generalizing to graphs with unbounded degrees sounds not practically important when verifying small graphs is already hard.
> > >
> > > Overall, there are theoretical merits in this paper and the theories may interest some people. But I think they are only for a minor problem that is not useful in practice, and the contribution is not significant enough. I would update my score to 5 accordingly.

---

> > > > ### Author Response · Authors · 2022-11-18
> > > > **response to your adjustments to your initial review and remaining concerns**
> > > >
> > > > **First, thanks for continuing the discussion. We also strongly appreciate your willingness to reconsider your original rating and improve it accordingly.**
> > > >
> > > > However, we do not agree with your final description, evaluating our contributions as *"not useful in practice, and ... not significant enough"*, which ultimately results in your increased, but still below acceptance rating. In the following we comment on these matters.
> > > >
> > > > ---
> > > >
> > > > > *"I mean only proving that a verification is possible with a computationally infeasible construction does not make a sufficiently significant contribution, ..."*
> > > >
> > > > Here we agree with you. Taken on its own, Thm. 2 is an interesting contribution, but not significant enough to carry an entire paper. However, Thm. 2 does not stand alone: the contribution of our paper is first to show that decidability is not guaranteed in the context of formal verification of GNN (Thm. 1) and second to determine the boundary between decidability and undecidability (the interplay of Thm. 1 and Thm. 2). We also elaborated on this in the [second to last paragraph of a previous comment](https://openreview.net/forum?id=WlbG820mRH-&noteId=OZvrQYrhTRB). Additionally, we refer to Fig. 1 of our submission, which clarifies and visualizes the interplay of our separate results. Further below, we will again discuss why these results are also, and perhaps especially, of interest to practitioners.
> > > >
> > > > >*"This is a misunderstanding of my comments. I mean in practice, ..."*
> > > >
> > > > Unfortunately, we feel that the paper's results are continously misinterpreted by using their potential relevance in practice, that you judge as very low, as the only means to measure their contribution.
> > > > The CfP lists "theoretical issues in deep learning" as one of the relevant topics for ICLR. This is what this paper is about. Theory and practice are not two competing methodologies. They complement each other. It is not the task of theory to predict what can be done in practice, but - amongst others - to study fundamental limits.
> > > >
> > > > > *"... some constraints can naturally be added, including the size of the graph and degrees, and thus the decidability on graphs without these constraints is not important."*
> > > >
> > > > Thm. 1 says "Problem A is undecidable". Now, you argue "This doesn't matter. In practice, I can just use problem B instead." This may well be right, but it does not contradict Thm. 1, nor does it make Thm. 1 irrelevant, as it is an "upper bound" in the overall result of our work. We (again) give an intuitive explanation for this down below.
> > > >
> > > > > *"Overall, there are theoretical merits in this paper and the theories may interest some people. But I think they are only for a minor problem that is not useful in practice, and the contribution is not significant enough."*
> > > >
> > > > We appreciate that. We believe that we have clearly summarized the individual contributions of our work above and in earlier comments. Now, why are our overall results of interest to much, if not ultimately all, of those concerned with the design and use of (formal) verification algorithms of GNN (note: we covered this before [here](https://openreview.net/forum?id=WlbG820mRH-&noteId=2_6hlg4wF4E))? Our results are a signpost. They indicate that (sound and complete) verification of graph-classifier MPNN (and correspondingly similar models) is only possible if we somehow restrict the input space (Thm. 1). (We are aware that you still think that this is always the case in practice. We have responded to this opinion in detail [here](https://openreview.net/forum?id=WlbG820mRH-&noteId=2_6hlg4wF4E) and [here](https://openreview.net/forum?id=WlbG820mRH-&noteId=F4kat40Vglw)). Moreover, we have shown first ways to tweak parameters so that cound and complete verification becomes possible (Thm. 2), namely in the case of node-classifier MPNN by constraining the node degree. (How to make it efficient in practice is a different question.)
> > > > In summary, our work is a first step towards a clear understanding of what is possible or impossible in terms of formal verification of GNN.
> > > >
> > > > ---
> > > >
> > > > *Short general note: due to the length of the discussion, we tried to simplify our explanations and focus on the reviewer's concerns. Details such as the differences between adversarial robustness and output reachability or dependencies of different GNN models are also partially covered in our work, but do not seem to be relevant for this  discussion.*

---

> > > > > ### Comment · Reviewer_UR4T · 2022-11-19
> > > > > **About misunderstood review**
> > > > >
> > > > > >It's not a misunderstanding of comments. Unfortunately, you continue to misinterpret the paper's results by applying a narrow view on how to judge a paper's benefit, namely only when it does something "in practice". We give a general argument against such reasoning: the CfP lists "theoretical issues in deep learning" as one of the relevant topics for ICLR. This is what this paper is about. Your argumentation would render every paper that deals with theoretical issues only as non-acceptable, not based on its actual content but simply because it does not deal with something else. Theory and practice are not two competing methodologies. They complement each other. It is not the task of theory to predict what can be done in practice, but - amongst others - to study fundamental limits.
> > > > >
> > > > > **This is clearly misinterpreting my comments.** Of course theoretical deep learning is an important area in ICLR. Definitely I don’t mean theoretical papers are all non-acceptable. But a good paper is supposed to make significant contributions with practical implications and address important issues in the area. **I think this paper is still below the acceptance bar because I think the contributions in this paper are insufficient, not because this is purely a theory paper.**
> > > > >
> > > > > As I have said, NP-hardness in verification is known and is already fundamentally limiting the verification. Especially on verification for graphs with unbounded degree/size, NP-hardness is a stronger limit that can hinder exact verification in practice for unbounded cases. That’s also partly why I said verification for the unbounded case is not important as we have already known NP-hardness on verification so I would not pursue generalizing verification to the unbounded case in practice. Therefore, for now, the limit revealed by Theorem 1 (decidability) is not an important limit compared to NP-hardness done by previous works and the contribution is minor. However, it could potentially become more significant if the paper can prove that the verification is not NP-hard in the special case for MPNN considered in this paper, to eliminate the limit revealed in previous works so that decidability could really be interesting and important. Theorem 2 only showing the possibility without a computationally feasible method is not making a sufficient contribution either.

---

> > > > > > ### Author Response · Authors · 2022-11-19
> > > > > > **responding to your comments**
> > > > > >
> > > > > > We think the most important argument, which could help you to finally understand the contribution of our work, we have given [in our previous comment](https://openreview.net/forum?id=WlbG820mRH-&noteId=ItMs57VkckT) in the first and last paragraph. **In one sentence: Our results are an interplay (visualized by Fig. 1), which shows limits and possibilities in the area of formal verification of GNN. We have explained why we feel that "lack of practical relevance" misses the point of our work.**
> > > > > >
> > > > > > ---
> > > > > >
> > > > > > > *"This is clearly misinterpreting my comments....  Of course theoretical deep learning is an important area in ICLR. Definitely I don’t mean theoretical papers are all non-acceptable.*"
> > > > > >
> > > > > > With all due respect, that is what your comments come along as, since the relevance of the paper's results is debated based on the view that they are "not relevant in practice." We have repsonded to this several times in detail (see [this comment](https://openreview.net/forum?id=WlbG820mRH-&noteId=2_6hlg4wF4E), maybe [this comment](https://openreview.net/forum?id=WlbG820mRH-&noteId=F4kat40Vglw) to another reviewer and the [previous one](https://openreview.net/forum?id=WlbG820mRH-&noteId=ItMs57VkckT)) and made clear why it is 1. of theoretical interest ("decidability is not guaranteed") and 2. why it is even of high practical interest. Therefore we do not think there is anything more to gain here in this direction of the discussion.
> > > > > >
> > > > > > > *"As I have said, NP-hardness in verification is known and is already fundamentally limiting the verification."*
> > > > > >
> > > > > > This is a general, rather imprecise statements for which we are not given any evidence. We can only respond to this with a lot of interpretation.
> > > > > >
> > > > > > What do you mean by "verification" here? Are you referring to "verification of GNN"? If so, such results would clearly refute our work. But we know that they do not exist. So presumably you mean "verification of classical NN" again. We have argued several times why GNN and NN are different models (see the middle part [here](https://openreview.net/forum?id=WlbG820mRH-&noteId=VDN4408lpB)), why such comparisons are generally not correct (see our statement to the border between decidability and undecidability [here](https://openreview.net/forum?id=WlbG820mRH-&noteId=VDN4408lpB)) and what the problems with a translation from GNN to NN are (see [here](https://openreview.net/forum?id=WlbG820mRH-&noteId=VDN4408lpB)).
> > > > > >
> > > > > > > *"Therefore, for now, the limit revealed by Theorem 1 (decidability) is not an important limit compared to NP-hardness done by previous works ..."*
> > > > > >
> > > > > > We still assume that you refer to NP-hardness of "verifying classical NN". Please accept that this work does not deal with classical neural networks but with *graph* neural networks. One cannot simply translate these results to the GNN setting. We have already addressed this [here](https://openreview.net/forum?id=WlbG820mRH-&noteId=VDN4408lpB).
> > > > > >
> > > > > > > *"However, it could potentially become more significant if the paper can prove that the verification is not NP-hard in the special case for MPNN considered in this paper, ..."*
> > > > > >
> > > > > > We showed that the situation is "worse than NP-hard": for the graph-classifier MPNN setting we proved (in Thm. 1) that formal verification is impossible in the sense that corresponding decision problem is (not only NP-hard but in fact) undecidable.
> > > > > >
> > > > > > ---
> > > > > >
> > > > > > *Since this will probably be our last post before the rebuttal phase ends, we thank you for your willingness to continue the discussion.*

---

> > > > > > > ### Comment · Reviewer_UR4T · 2022-11-19
> > > > > > > **Disagree. Please make sure your claims are sufficiently supported.**
> > > > > > >
> > > > > > > >This is a good example of how you miss the overall implications of our work. We did exactly as you state. In particular, we showed that the situation is "worse than NP-hard": for the graph-classifier MPNN setting we proved that formal verification is impossible or, in other words, that the corresponding decision problem is undecidable. See Thm. 1.
> > > > > > >
> > > > > > > I disagree.
> > > > > > > Theorem 2 has shown that the undecidability can be bypassed simply by bounding the degree, even though this bound can be very large. But NP-hardness cannot be bypassed unless you make this bound small enough. Clearly the undecidability is not worse than NP-hardness.
> > > > > > > Please do not claim such a "vague" and "baseless" `worse than NP-hard` without enough evidence.
> > > > > > >
> > > > > > > **You are submitting this paper so you are responsible for sufficiently supporting your claims.** Given the existence of previous works on classifical NN, you need to justify that the NP-hardness does not apply in your case, otherwise as I said before, the undeciability is not meaningful. You can't keep ignoring existing works on classifical and more fundamental NNs only because you're now working on GNN.

---

> > > > > > > > ### Comment · Reviewer_UR4T · 2022-11-19
> > > > > > > > **Failure to support "worse than NP-hard"**
> > > > > > > >
> > > > > > > > Actually the failure to support that decidability is "worse than NP-hard" poses a significant weakness of this paper. This paper provides no evidence to support such a claim. Without it, given the existing works on NP-hardness, showing the undecidability is insufficient to support the significance of this work. See my last thread on why I disagree decidability is worse than NP-hard.
> > > > > > > >
> > > > > > > > The authors refuse to compare with existing works on regular NN, by saying this paper works on GNN but not regular NN. Please note that GNN is an extension of regular NN. There are works about NP-hardness of verification on regular NN and you need to show that why their results don't apply to you in a formal way, e.g., by proving that the verification in your case is not NP-hard, which is missing in this paper and I am not sure if it is actually possible to prove that.
> > > > > > > >
> > > > > > > > Since the authors complain about "general, imprecise statements without any evidence", I would list some references:
> > > > > > > >
> > > > > > > > - Katz, G., Barrett, C., Dill, D. L., Julian, K., & Kochenderfer, M. J. (2021). Reluplex: a calculus for reasoning about deep neural networks. Formal Methods in System Design, 1-30.
> > > > > > > >
> > > > > > > > - Sälzer, Marco, and Martin Lange. "Reachability is NP-Complete Even for the Simplest Neural Networks." International Conference on Reachability Problems. Springer, Cham, 2021.
> > > > > > > >
> > > > > > > > I would say **decidability would not limit NN verification more than NP-hardness and likely much less than NP-hardness in practice**. If this is true, the results in this paper are not useful given existing works on NP-hardness. The authors are responsible to provide evidence on why it is not the case here. I sincerely ask the authors not to evade this issue and ignore previous works by saying GNN is not classifical NN.

---

> > > > > > > > > ### Author Response · Authors · 2022-11-19
> > > > > > > > > **we fear that the discussion is going out of hand**
> > > > > > > > >
> > > > > > > > > *This comment refers to the one above and its [previous one](https://openreview.net/forum?id=WlbG820mRH-&noteId=u3TrgCTLzp).*
> > > > > > > > >
> > > > > > > > > ---
> > > > > > > > >
> > > > > > > > > >*"Please do not claim such a "vague" and "baseless" worse than NP-hard without enough evidence."*
> > > > > > > > >
> > > > > > > > > We have deliberately put the statement you are referring to in quotation marks because it is obviously to be understood informally. The comparison was between an undecidable problem and an NP-hard but decidable problem. From a computability point of view, an unsolvable problem is considered to be "worse" than a solvable one.
> > > > > > > > >
> > > > > > > > > > *"Please note that GNN is an extension of regular NN."*
> > > > > > > > >
> > > > > > > > > This statement oversimplifies the situation. GNN are models which calculate functions over graphs. The connection to NN comes from the fact that parts of this computation depend on learnable parameters, which in usual applications are represented by classical NN.
> > > > > > > > >
> > > > > > > > > Another example which shows that extensions of a model do not necessarily lead to straight-forward extensions of results: linear programming (LP) can be seen as an extension of integer linear programming (ILP) - since every integer is also a real number - but LP is in P whereas ILP is NP-hard.
> > > > > > > > >
> > > > > > > > > > *"Since the authors complain about "general, imprecise statements without any evidence", I would list some references:"*
> > > > > > > > >
> > > > > > > > > We are aware of the works you are referring to. Both were cited in our original submission already. Since GNN are not the same as NN, these results do not apply directly. We explained this in our previous arguments.
> > > > > > > > >
> > > > > > > > > ---
> > > > > > > > >
> > > > > > > > > *Although we are not entirely happy with the route that this conversation is taking, we thank you for your active participation in the rebuttal phase.*

---

> > > > > > > > > > ### Comment · Reviewer_UR4T · 2022-11-19
> > > > > > > > > > **Reply**
> > > > > > > > > >
> > > > > > > > > > >We have deliberately put the statement you are referring to in quotation marks because it is obviously to be understood informally. The comparison was between an undecidable problem and an NP-hard but decidable problem. We think it is common sense to see that , from a computability point of view, an unsolvable problem is "worse" than a solvable one.
> > > > > > > > > >
> > > > > > > > > > The issue is that the undecidability exists only when the degree is unbounded. It immediately becomes decidable once you add a bound on the degree (even though this bound can be very large), as you show in Theorem 2. I believe NP-hardness is stronger in this case, which even restricts the bound from being large in practice, for computational feasibility. It is also why Theorem 1 is not practically meaningful because its results cannot imply but are actually opposite to the actual issue in practice (Theorem 2).
> > > > > > > > > >
> > > > > > > > > > >This statement oversimplifies the situation. GNN are models which calculate functions over graphs. The connection to NN comes from the fact that parts of this computation depend on learnable parameters, which in usual applications are represented by classical NN. Sorry to say this, but we have come to feel that your criticism arises from a lack of understanding about the subject.
> > > > > > > > > >
> > > > > > > > > > I clearly know GNN myself… The issue is the authors are refusing to consider existing works on NP-hardness which is well known in the NN verification area so the authors are not supposed to ignore the issue. Of course I acknowledge GNN does not equal to classical NN, but this doesn’t immediately invalidate existing works on NP-hardness unless you provide evidence that the problem here is not NP-hard. I didn't say you didn't cite the works I listed, but I mean you need to address the issue why decidability is harder than NP-hardness in MPNN.
> > > > > > > > > >
> > > > > > > > > > It seems that this reply has been edited. I would also clarify that why you don’t see the separate comment previously posted by me on OpenReview. 1) I posted that comment first and used its content to update my review, not I copied my review into a separate comment; 2) I didn’t delete it but just changed its visibility, to match the visibility of replies by other reviewers.

---

> > > > > > > > > > > ### Author Response · Authors · 2022-11-20
> > > > > > > > > > > **On undecidability vs. NP-hardness**
> > > > > > > > > > >
> > > > > > > > > > > > It immediately becomes decidable once you add a bound on the degree (even though this bound can be very large), as you show in Theorem 2. I believe NP-hardness is stronger in this case, which even restricts the bound from being large in practice, for computational feasibility.
> > > > > > > > > > >
> > > > > > > > > > > It follows directly from the definitions of undecidability (i.e. $\Sigma^0_1$-hardness or $\Pi^0_1$-hardness) that any undecidable problem is likely to be NP-hard or coNP-hard (if the reduction from the (non-)halting problem is polynomial, which is often the case). Moreover, the other direction does not hold as there are many problems that are NP-hard (or coNP-hard) that are not undecidable. Hence, undecidability is the (much) stronger statement, and this is usually not up for debate.

---

> > > > > > > > > > > > ### Comment · Reviewer_UR4T · 2022-11-20
> > > > > > > > > > > > **Conditions cannot be ignored.**
> > > > > > > > > > > >
> > > > > > > > > > > > The comparison is meaningless if the conditions for undecidability and NP-hardness are ignored. Note that unlike NP-hardness, undecidability in this paper only holds when degrees are unbounded.
> > > > > > > > > > > >
> > > > > > > > > > > > NP-hardness is practically meaningful for NN verification because it implies the size of the problem cannot be arbitrarily large and should be sufficiently small for computationally feasibility.
> > > > > > > > > > > >
> > > > > > > > > > > > Undecidability in this paper is not practically meaningful for now, because the unbounded case is coverd by NP-hardness which already implies the infeasibility in practice, while it still does not hinder verification for the bounded cases.

---

> ### Author Response · Authors · 2022-11-19
> **final comment on ... (deleted)**
>
> *The discussion with reviewer UR4T has become notably long. It started [here](https://openreview.net/forum?id=WlbG820mRH-&noteId=6fe_RS3HQrC). Reviewer UR4T has deleted most parts of his/her former review which makes it hard to follow the discussion in the beginning.*
>
> **We do not appreciate about certain changes in the discussion that made it hard for us to respond. Here is a brief summary of the process:**
>
> - The reviewers original review was from October 14, which was substantially revised on November 06, presumably after seeing that the opinions of all the other reviewers is different. This made it hard to give early comments regarding UR4T's concerns. At this point UR4T's rating of our submission was '3'.
>
> - Then reviewer UR4T made another general comment (which is deleted now) which essentially was a copy of the review as it was present up to this morning.
>
> - After some reasonable discussion, reviewer UR4T changed the rating to '5' (see comment [here](https://openreview.net/forum?id=WlbG820mRH-&noteId=CyL5NLyjmws)).
>
> - In the continuing discussion UR4T brought up reasons (we shortly comment on these below) for why our work is not relevant which we tried to respond to accordingly. This ended in an again revised review, in which the rating was changed back to '3'.
>
> **The process of repeatedly revising reviews, including the up- and downgrading, is - in our view - in contrast to UR4T's confidence of '4'.**
>
> We would not normally bring this up as we do not feel that this is a necessarily a matter of rebuttal, but reviewer UR4T has pointed out the confidence rating of the other reviews.
>
> Now for the new reason why UR4T finds our work unacceptable:
> > *"Actually the failure to support that decidability is "worse than NP-hard" (unconvincingly claimed by the authors) poses a significant weakness of this paper. "*
>
> This is not true. We said in [this comment](https://openreview.net/forum?id=WlbG820mRH-&noteId=LXeVO_iFIJ) that *undecidability* is clearly "worse than NP-hard" (note: this referred to an NP-hard problem which is decidable). We explained this informal statement in detail [here](https://openreview.net/forum?id=WlbG820mRH-&noteId=RdgusvE3m0).
>
> **We fear that any continuation of the discussion will lead to further changes in the evaluation of our work that are hard for us to comprehend. Therefore, we would like stop the discussion at this point and ask for further edits and deletions of existing comments not to be made anymore, such that traceability of the discussion is maintained.**

---

> ### Author Response · Authors · 2022-11-23
> **reset**
>
> Dear reviewer,
>
> I assume we can agree that the discussion has gone a little bit out of hand. This is an attempt to take it back to a more sound level of respect and rationality.
>
> We, on our side, had a few days where we could not communicate, so many of the comments made in the last days could not go through the internal filters first. This was also not helped by the fact that we were under the impression that the rebuttal phase would close on Saturday, so some of those comments have also been rushed in the light of this assumed deadline for responses.
>
> I have now taken the liberty to go through all our comments and to eliminate formulations that could well be interpreted as accusatory or disrespectful. There are of course original formulations left in your quotations, but that's as may be. It is not an attempt to eradicate what has been said, but only a sign of good will to come to some conclusion here that contains some mutual understanding about the paper.
>
> I think the points have been made clear that you debate the practical relevance of the paper's results, and that we consider this to be missing the point of the contribution. We are happy to leave it like this (in the sense of the well-known "agree to disagree"). Instead, I'd like to suggest that we go straight to the ultimate point of rebuttal, if you're ok with that:
>
> There are relatively high stakes for us in the game; we submitted a theory paper to a very reputable machine learning conference whose CfP welcomes such contributions, and there is not exactly an abundance of these. The paper studies fundamental limits of GNN which is novel and highly relevant *in theory at least*, as it starts to pinpoint the location of the GNN model(s) in the landscape of expressiveness and computational complexity. The paper received marks of 8,10, 8 and 3. The first three are highly encouraging for us and indicate to us that we are not completely wrong with what we are doing here. After some discussions and clarifications with you, you raised your mark to 5 which - so it has to be said - we appreciated a lot. Then the discussion continued and got slightly out of hand, and you changed your mark back to 3 which, clearly, did not make us particularly happy. You are of course entirely free in your judgment but this is rebuttal, so we would like to understand what caused this second change of mind, as we are of course trying to refute criticism or use it for improvements of the paper at hand. So  something must have happened because of which you then ranked the paper worse than you did in between.
>
> - If it is the tone that the comments have taken then we do apologise again, and, as said above,  have edited them accordingly in order not to let the tone dominate the discussion instead of the validity of the arguments.
>
> - If there are additional concerns of yours about the paper's quality that have only been uncovered in this discussion then we are happy to try to refute them (but it might be necessary to restate them clearly and succinctly).
>   - We are happy to accept your judgment about practical relevance and would like the paper to be seen as a contribution expanding the theory of expressiveness and computability for machine learning models.
>   - We have contacted the authors of one of the paper you mentioned (and we cited) about NNs and NP-completeness, and they confirmed that their findings do not cover the case of GNN models.
>
> In clear words: we would of course like to get the best marks possible for our paper. What is it that you would like to have here?

---

### Official Review · Reviewer_8dRF · 2022-10-21

**Confidence:** 3
**Correctness:** 4
**Technical Novelty And Significance:** 4
**Empirical Novelty And Significance:** Not applicable
**Recommendation:** 8

**Clarity, Quality, Novelty And Reproducibility:**

The work does not contain any experimental elements, so there is no consideration of reproducibility.
The proof of the paper in the appendix are self-contained, so the arguments can be independently checked. To the best of my knowledge, the results in this work are novel.


## Minor typos:
- In the first half of page 2 -> "(in-)valud output"
- In proof of Lemma 2. -> "for each for each"
- In proof of Theorem 4. -> In the definition of $\langle x = m \rangle$, should it be an abbreviation of $\langle -x \leq -m\rangle + \langle x \leq m \rangle$?
- Appendix B.2 -> "Reducable" -> "Reducible"?

**Strength And Weaknesses:**

The paper is very clearly written, with the authors having made a clear effort to be rigorous and provide definitions for all the constructions that they use. There is a clear structure to the paper, with the results being announced first, the strategy of the proof being presented and then executed, with the detailed version being present in the appendix.
The proofs are decomposed into logical parts, making it possible to get relatively quickly an intuition for the reduction being operated.

In addition, the paper is quite clear in Section 6 to delineate the settings in which their results are applicable and the ones in which they are not.

**Summary Of The Paper:**

This paper presents theoretical results about the decidability of verification problems of Message Passing Neural Networks, which are a type of graph neural network.
Graph neural network are a flexible deep learning architecture that operate on graph data, which can be of variable sizes.

The first main result of the paper is in Theorem 1, proving that if the input specification is permissive enough (allowing unbounded size of input graphs and without imposing restrictions on node degrees and labels), then it is not possible to perform reachability verification (answer the question "is there a graph such that a certain output will be produced by the MPNN when evaluated over this graph?").
The proof is performed by reducing a known undecidable problem (Post's correspondence Problem) to a graph satisfiability problem based on linear constraints ( in section 4.1) and then, reducing that graph satisfiability problem to a MPNN verification problem (in section 4.2).

The other main result  is Theorem 2 which shows that if the input graph have bounded degrees, then the verification problem become decidable. The proof rely on the fact that due to the way MPNN work (each node can only process information that is most $\texttt{GNN depth}$ nodes away, then if a graph satisfy a feasibility problem, there must a tree that satisfy it as well. Due to the assumption of bounded degrees, only a finite amount of those trees can exist, so it's possible to enumerate them, and to perform verification independently on each tree using traditional verification technique.

**Summary Of The Review:**

This is a completely theoretical paper that proves results on the limitations of verifiability of graph neural network.
This is useful as it tells us that there is no point in researching verification algorithms that would be applicable to the most general settings, as these problems are not going to be decidable. The whole paper is self contained and clearly written, so I recommend acceptance, even if this does not inform real applications (where we could always consider that the graph considered would be bounded).

---

> ### Author Response · Authors · 2022-11-12
> **response regarding the practical applicability of our result(s)**
>
> First of all, thank you for your thorough reading and reviewing of our submitted paper.
> Special thanks also for the smaller hints regarding typos etc.
>
> > Minor Parts and Typos
>
> We fixed all typos and and minor issues that you spotted. See the revised version of our paper.
> Thanks again for pointing these out!
>
>
>
> > ... even if this does not inform real applications (where we could always consider that the graph considered would be bounded).
>
> Thanks for this remark, allowing us to further elaborate on this. The point here is the subtle difference between "being bounded", meaning "$ \leq n $ for a
> concrete, given natural number $n$", and just "being finite" meaning "$\leq n$ for some
> arbitrary $n$".
> The size of each graph encountered in the forseen practical contexts is implicitly
> bounded by the fact that every implementation of an algorithm will only every be
> applied to finitely many inputs in its lifetime, or it is run on a machine whose
> memory will be exhausted for graphs of too large size. Hence, the maximum amongst
> all inputs occurring after all is a possible bound. Likewise, a bound could of course
> be introduced explicitly in an algorithm.
> In both cases, any algorithm that presupposes such a bound is limited in comparison
> with one that does not presuppose such a bound. Undecidability results like the one
> in Th. 1 are of course stronger when they cover not only the limited algorithms. This
> is why it makes sense to not assume boundedness (because the introduction of a bound
> would then only cover those practical settings that are bounded in this particular
> way, and others would not be covered) but instead "only" finiteness, because this then
> covers *all* settings for any arbitrary bound.
> So we would argue that this does indeed inform practical applications, namely in the
> following sense. As a consequence of Th. 1, in GNN verification one needs to either
> A) give up termination, soundness or completeness of the verification procedure, or
> B) make limiting assumptions on the gaphs under consideration (like degree bounds).
> While the latter may be done regularly in practice, this does not contradict Th. 1
> as Th. 1 makes a statement about possible alternatives, namely: if your verification
> procedure does not make such a priori assumptions, then it cannot be sound, complete
> and terminating.

---

### Official Review · Reviewer_LFLB · 2022-10-24

**Confidence:** 3
**Correctness:** 4
**Technical Novelty And Significance:** 4
**Empirical Novelty And Significance:** Not applicable
**Recommendation:** 10

**Clarity, Quality, Novelty And Reproducibility:**

This paper was very well written and I was able to follow the proof sketches. If there is an error, i believe it is either subtle or in the full proof which was not fully checked.

As stated above the work is very important and to my knowledge novel.

I would perhaps ask for another pass on the PCP exposition, in particular making figure 2 contain the set of tiles next to the concatenation shown.

**Strength And Weaknesses:**

# Strengths

1. The topic of verification is a very important problem, particularly as such systems get deployed in more safety critical systems. Knowing the limitations of verification is critical to carving out tractable subsets of the problem space. Theoretical treatment of such topics typically lags behind the development of the models, and so this is welcomed progress.

2. The provided reduction is succinct and fairly easy to follow, although I must admit to needing a refresher on PCP.

# Weaknesses

1. The paper focuses on a fairly small (but as far as I know practically used) subset of MPNNs. There is a discussion that the specifics could be relaxed, but many alternative formulations exist, e.g., a different aggregation function.

**Summary Of The Paper:**

This problem considers the question of whether or not it is possible to verify robustness and output reachability properties of message passing neural network (MPNN), e.g. graph neural networks. The paper proves via a succinct reduction to Post's Correspondence Problem that verification of such properties on large classes on MPNNs is undecidable. Furthermore, the illustrate that limiting the class of input graphs, e.g., limiting the degree of the nodes, is sufficient to make the problem decidable. This generalizes the feasibility established by explicit algorithms appearing in the literature.

**Summary Of The Review:**

Overall, I think this i an excellent contribution to the theoretical understanding of message passing and the limits on verification.

The main take aways are that:

1. arbitrary graphs are too expressive.
2. reasonable restrictions on node degree is sufficient.

---

> ### Author Response · Authors · 2022-11-12
> **response regarding your reference to the choice of considered model(s)**
>
> Thank you very much for your thorough review and comments!
>
>
> > The paper focuses on a fairly small (but as far as I know practically used) subset of MPNNs. There is a discussion that the specifics could be relaxed, but many alternative formulations exist, e.g., a
> different aggregation function.
>
> We agree, there are many models in the GNN framework for which similar investigations would be interesting, to get a full picture of the broad landscape of (formal) GNN verification.
> Our work should be seen as a first, fundamental step in this direction, as it covers the popular MPNN model in a common form. Since the scope of a concise paper is limited, we refer to future work.
> (There is also a closely related comment of ours in the response to reviewer 5S64.)

---

### Official Review · Reviewer_5S64 · 2022-10-25

**Confidence:** 2
**Correctness:** 4
**Technical Novelty And Significance:** 4
**Empirical Novelty And Significance:** Not applicable
**Recommendation:** 8

**Clarity, Quality, Novelty And Reproducibility:**

I think that the writing is overall clear and crisp. I did not spot any major problem in the formal proofs, the high-level approach and reductions make sense to me but I couldn't carefully check the details in the appendix. I think that the paper could be improved as follows.

---

**The presentation could be improved**

I understand that the main text is meant to provide intuitions and key ideas behind the theoretical results, whose formal treatment must be deferred to the appendix.

Nonetheless, I think that Section 3 would be much more accessible if the input/output specifications in Fig. 1 were introduced at the beginning, maybe with some examples on what they entail.

---

While the relation between ORP and ARP is clear, I wish that the relation between the graph and node classification task was discussed, connecting the negative (Th/Cor. 1) and positive (Th/Cor. 2) results.


---

    "[..] where φ discr is built like above [..]"

That formula is not defined above.

---

**It is not clear how realistic are the setting and assumptions made by the authors**

    "In this paper, we only consider MPNN where the aggregation, combination and readout parts are given as follows"

How limiting and how realistic is this restriction described here? Even if an in-depth discussion is deferred to Section 6, it would be good to motivate it, possibly citing multiple influential works that fall into this category.

---

**Minors**


    "[..] most work considers GNN used for node-classification and
    among such most common are edge modifications of a fixed input
    graph [..]"

Most common *attacks*?

---

    "Using this lemma, in order to prove Theorem 1, it suffices to show that GCP is not formally verifiable"

that GCP is *undecidable*.

---

    "[..] - here: words witnessing a solution of a PCP instance – by means of
    vectors and the operations that can be carried out on them inside a MPNN."

I don't understand this sentence.

---

    "To keep the notation clear we write – if unambiguous – we denote some variable x i in node and graph conditions by its index i."

Typo?


**Strength And Weaknesses:**

Strengths:
- The paper contributes to a fundamental problem in the context of safe/verifiable AI for a popular class of ML models
- The theoretical groundwork provided in the paper, such as the connection between formal verification of GNNs and the newly introduced Graph Linear Programming (GLP), could benefit future efforts in this area.

Weaknesses:
- The presentation could be improved in some parts
- It is not clear to me how realistic are the setting and assumptions made by the authors

**Summary Of The Paper:**

This paper provides theoretical results on the formal verifiability of message passing neural networks (MPNNs), a popular class of graph neural network models, for graph and node classification tasks.
In particular, it proves that, without assuming a bound on the degree of the graph, the formal verification of output reachability and adversarial robustness is undecidable for graph classification problems. At the same time, the paper provides positive results by showing that bounding the degree makes the verification of node classifiers decidable.



**Summary Of The Review:**

The paper contributes with some key theoretical results on the problem of verifying MPNNs, as well as providing useful groundwork for future efforts in the area. I think that the problem considered in this paper is very relevant. The overall methodology seems correct and, to the best of my knowledge, novel. However, I couldn't verify the correctness of the proofs in the appendix. The presentation could be improved in some parts, but overall I think that the authors did a good job in clearly presenting the key concepts and intuitions.

---

> ### Author Response · Authors · 2022-11-12
> **response regarding your references to weaknesses and minor errors**
>
> First of all, thank you for taking the time to do a thorough review,
> which even includes minor comments on notation and spelling.
>
> > Minor Parts, Typos
>
> We have considered and corrected  all of the "minor comments." See the
> corresponding revised version of the paper. Thanks again, for pointing these out.
>
>
> >The presentation could be improved
>
> On your question regarding graph vs node-classifier: the essential difference for our
> results is that graph-classifier MPNN can make classification decision based on global information
> (due to the sum in the readout layer), which seems to be needed for the proof of Th.1. Node-classifier MPNN,
> on the other hand, can only use local information (around the target node) for their decision.
> One can circumvent this "limitation" of node-classifying MPNN by adding a master node to the input graph, connected to all other nodes, and considering this as the target node. However, adding such master nodes leads to graph classes with unbounded degree. This indirectly implies why Th.2 holds if we bound the degree of graphs: master nodes are prohibited. We shortly discussed this in Sect. 6.
>
> We agree with you that broader discussions and explanations in the (main) paper are desirable. However, due to the page limit
> we would need to shorten other parts, which is not possible without losing needed proofs, intuitions etc.
>
>
> >It is not clear how realistic the attitudes and assumptions made by the
> authors are.
>
> This is of course an important point. As you stated, we discussed this problem in Section 6:
> Our results apply without further ado only to GNN models with higher (Th.1) / lower (Th.2) expressive
> power as the ones considered. Currently, the GNN framework seems to be very heterogeneous.
> For example, Wu et al. [1] categorize the different models as RecGNN,
> spectral GN, spatial GNN, and possible hybrids. By choosing MPNN, we
> have taken a blueprint of the spatial GNN category, which is undoubtedly
> the most considered category at present.
>
> However, MPNNs are not
> concretely defined, as the specific form of the aggregation,
> combination, and selection parts is not specified by the model.
> A common choice is to represent these parts by simple sums (aggregation part)
> or classical NNs with few layers (combination and readout parts).
> Because of your hint and to underline the commonality of our choice, we added some citations of foundational work/surveys in Sect. 2. See the revised version of the paper. (Unfortunately, we do not know of any work that empirically confirms the commonality of this choice. Such a work would definitely be an important contribution to better understand the "GNN Zoo".)
> Focusing on such common models increases the general applicability of our results. However, we agree that there are open questions regarding different models and specifications, which must be answered in future work.
>
> *[1] Zonghan Wu, Shirui Pan, Fengwen Chen, Guodong Long, Chengqi Zhang, and Philip S. Yu. A comprehensive survey on graph neural networks. IEEE Trans. Neural Networks Learn. Syst., 32 (1):4-24, 2021. doi: 10.1109*

---

> > ### Comment · Reviewer_5S64 · 2022-11-16
> > **Response to the authors**
> >
> > > On your question regarding graph vs node-classifier: the essential difference for our results is that graph-classifier MPNN can make classification decision based on global information (due to the sum in the readout layer), which seems to be needed for the proof of Th.1. Node-classifier MPNN, on the other hand, can only use local information (around the target node) for their decision. **One can circumvent this "limitation" of node-classifying MPNN by adding a master node to the input graph, connected to all other nodes, and considering this as the target node. However, adding such master nodes leads to graph classes with unbounded degree.** This indirectly implies why Th.2 holds if we bound the degree of graphs: master nodes are prohibited.
> >
> > Thank you for your response. The sentence in bold made it click for me.. I would mention this intuition earlier in the text, maybe when you describe Fig.1 or right after the theorems.

---

> > > ### Author Response · Authors · 2022-11-18
> > > **thanks for the suggestion**
> > >
> > > We are glad that we could help you with this explanation
> > >
> > > > I would mention this intuition earlier in the text, maybe when you describe Fig.1 or right after the theorems.
> > >
> > > Thats a great suggestion. We moved the explanation, formerly present in Sect. 6, up to the end of Sect. 3. See the revised version of our submission. Thank you for the tip!

---

### Author Response · Authors · 2022-12-12
**summary of the rebuttal phase for Paper4236 (from the authors perspective)**

**We think it will help reproducibility of the rebuttal phase discussion(s) if we briefly summarize the individual discussion threads.**

---

### Discussion with reviewer 5S64 (original rating: 8; after rebuttal phase: 8)

As the original and unmodified rating attests, reviewer 5S64 sees our contributions as relevant and valuable. We refer to the original [review](https://openreview.net/forum?id=WlbG820mRH-&noteId=YAFIzyeT4mh).

Reviewer 5S64 noted two weaknesses in our work: (1) improvement in presentation and (2) relevance of the specific setting. We incorporated all of the specific comments on (1) into our revised version of the paper. Regarding (2), we have further elaborated on the relevance of our setting [here](https://openreview.net/forum?id=WlbG820mRH-&noteId=d2NCG8YfoX) and provided supporting citations in the revised version.

---

### Discussion with reviewer LFLB (original rating: 10; after rebuttal phase: 10)

As the original and unmodified rating attests, reviewer LFLB sees our submission as a highly important contribution. We refer to the original [review](https://openreview.net/forum?id=WlbG820mRH-&noteId=qwcodnsrZK).

Reviewer LFLB noted one weakness: the results of our work apply without further ado to only a small subset of the models in the GNN framework. We explained our choice of models [here](https://openreview.net/forum?id=WlbG820mRH-&noteId=1WP_9bmna3) and also in a similar explanation to reviewer 5S64 [here](https://openreview.net/forum?id=WlbG820mRH-&noteId=d2NCG8YfoX).

---

### Discussion with reviewer 8dRF (original rating: 8; after rebuttal phase: 8)

As the original and unmodified rating attests, reviewer 5S64 sees our contributions as relevant and valuable. We refer to the original [review](https://openreview.net/forum?id=WlbG820mRH-&noteId=ReY-V20A_v).

Reviewer 8dRF noted one weakness: our results are only applicable for the most general setting and, thus, do not inform practical applications. We explained our differing view on this matter [here](https://openreview.net/forum?id=WlbG820mRH-&noteId=F4kat40Vglw). We also addressed this matter [here](https://openreview.net/forum?id=WlbG820mRH-&noteId=2_6hlg4wF4E).

---

### Discussion with reviewer UR4T (original rating: 3; after rebuttal phase: 3 (with an intermediate change to 5))

As the original and ultimate rating attests, reviewer UR4T does not see our contribution as valuable.

The discussion with reviewer UR4T is hard to follow, as the reviewer has made several extensive revisions of the original review, sometimes put responses in odd places in the discussion threads and removed them afterwards. We summarized these problems [here](https://openreview.net/forum?id=WlbG820mRH-&noteId=vUPcMw957gS).

The original main concerns of reviewer UR4T were: (1) our main impossibility result (Thm. 1) is not meaningful for practitioners, as the scenario is artificial and (2) our main possibility results (Thm. 2) is meaningless, as there are stronger results in the area of classical neural network verification. Regarding (1), we made an extensive comment [here](https://openreview.net/forum?id=WlbG820mRH-&noteId=2_6hlg4wF4E) explaining the value of Thm. 1 to the GNN verification community. Regarding (2), we made a first comment [here](https://openreview.net/forum?id=WlbG820mRH-&noteId=OZvrQYrhTRB), explaining that classical neural network and GNN, which this work addresses, are separate models. Claims (1) and (2) were also the overall subject of the continuing discussion starting [here](https://openreview.net/forum?id=WlbG820mRH-&noteId=VDN4408lpB), discussed from different points of view.

In a following [comment](https://openreview.net/forum?id=WlbG820mRH-&noteId=CyL5NLyjmws), reviewer UR4T admitted theoretical merits of our work and raised the rating to 5. However, in the continuing discussion reviewer UR4T expressed another concern: (3) our work failed to support the claim (informally made by us [here](https://openreview.net/forum?id=WlbG820mRH-&noteId=LXeVO_iFIJ)) that undecidability is worse than NP-hard. This lead to a revision of reviewer UR4T’s review (its current form) and a change of the rating back to 3.
We addressed concern (3) [here](https://openreview.net/forum?id=WlbG820mRH-&noteId=RdgusvE3m0) and [here](https://openreview.net/forum?id=WlbG820mRH-&noteId=zLwBkAxUOE6) giving general counterarguments. In the continuing discussion UR4T did not accept these these arguments by referring to (1) again. We felt that the discussion was stuck and derailed and ended it temporarily [here](https://openreview.net/forum?id=WlbG820mRH-&noteId=vUPcMw957gS). Later, we tried to resume it [here](https://openreview.net/forum?id=WlbG820mRH-&noteId=YORZrqiOHdI) by taking some of the heat out of it, but reviewer UR4T did not respond anymore.

---

### Decision · Program_Chairs · 2023-01-20

**Decision:**

Accept: poster

**Justification For Why Not Higher Score:**

While the authors establish an important theoretical limitation of graph neural network verification, the significance of this result to practical verification problems is not clear since in many applications an assumption of bounded degree suffices.

**Justification For Why Not Lower Score:**

The authors establish a stronger theoretical limit on verifiability of graph neural networks than any previous work.

**Metareview: Summary, Strengths And Weaknesses:**

The authors prove fundamental limits on the ability to formally verify graph neural networks. In particular, they prove that the neural network verification problem is undecidable when the degree of the graph is unbounded.

Strengths:
1. The authors establish a strong fundamental limitation to formally verifying graph neural networks.
2. The proof techniques used by the authors to establish undecidability are novel and could be of independent interest to the community

Weaknesses:
1. The limitations established by the authors only apply to graphs where the degree can grow unboundedly, and still allows for algorithms that assume an a-priori bound on the degree of the graph.

**Note From Pc:**

if the above contains the word "oral" or "spotlight" please see: "oral" presentation means -> notable-top-5% and "spotlight" means -> notable-top-25%. As stated in our emails, we are disassociating presentation type from AC recommendations

**Summary Of Ac-Reviewer Meeting:**

The primary outcome of the discussion was to agree on the reasons to accept/reject. All reviewers except one were in favour of acceptance, and the key objection of the reviewer was that the result only applies to graphs of unbounded degree. As an AC, I made the determination that while this limitation is important for the authors to highlight, the paper still establishes a fundamental result on undecidability of formal verifiability of graph neural networks. Summary of discussion:

Reasons favoring acceptance:

Undecidability is a much stronger limitation than NP-hardness. Many NP-hard problems are solved in practice but undecidability has a fundamental limitation.

Even though the problem is decidable given an a-priori bound on the degree, in many practical situations, such a bound may not be available a-priori, leading to the algorithm designer having to make a conservative overestimate of the degree bound (hence impacting the computational complexity) or redesign the algorithm when the deployment encounters a graph of higher degree than the a-prior bound.

The proof techniques themselves involve interesting theoretical constructions that could be of independent value.

Reasons favoring rejection

Undecidability of graphs with unbounded degree is not important, because we know that verification is NP-hard and therefore any problems with too large a size will be be impossible to solve in practice anyway